# MEnvAgent: Scalable Polyglot Environment Construction for Verifiable Software Engineering

**Chuanzhe Guo** [* 1 2]   **Jingjing Wu** [* 2]   **Sijun He** [2]   **Yang Chen** [2]   **Zhaoqi Kuang** [2]   **Shilong Fan** [2]   **Bingjin Chen** [2]
**Siqi Bao** [† 2]   **Jing Liu** [2]   **Hua Wu** [2]   **Qingfu Zhu** [1]   **Wanxiang Che** [† 1]   **Haifeng Wang** [2]

## Abstract

The evolution of Large Language Model (LLM) agents for software engineering (SWE) is constrained by the scarcity of verifiable datasets, a bottleneck stemming from the complexity of constructing executable environments across diverse languages. To address this, we introduce **MEnvAgent**, a **M**ulti-language framework for automated **Env**ironment construction that facilitates scalable generation of verifiable task instances. MEnvAgent employs a multi-agent Planning-Execution-Verification architecture to autonomously resolve construction failures and integrates a novel Environment Reuse Mechanism that reduces computational overhead by incrementally patching historical environments. Evaluations on MEnvBench, a new benchmark comprising 1,000 tasks across 10 languages, demonstrate that MEnvAgent outperforms baselines, improving Fail-to-Pass (F2P) rates by **8.6%** while reducing time costs by **43%**. Additionally, we demonstrate the utility of MEnvAgent by constructing MEnvData-SWE, the largest open-source polyglot dataset of realistic verifiable Docker environments to date, alongside solution trajectories that enable consistent performance gains on SWE tasks across a wide range of models.

## 1. Introduction

The rapid evolution of Large Language Models (LLMs) has significantly advanced the exploration of repository-level code modification tasks within software engineering. Real-world issue resolution benchmarks, such as SWE-bench

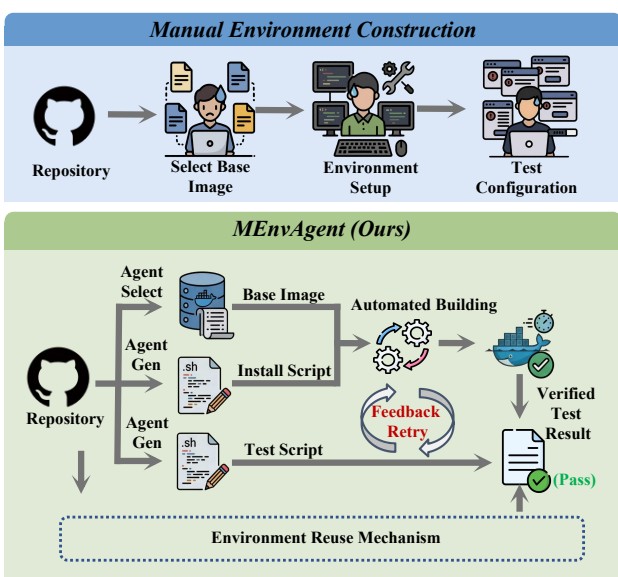

*Figure 1.* **Comparison between manual environment construction and MEnvAgent (Ours).** MEnvAgent leverages multi-agent collaboration to achieve automated environment construction, characterized by an efficient environment reuse mechanism.

and its variants (Jimenez et al., 2024; Yang et al., 2025c; Zan et al., 2025), have emerged as the standard for evaluating the coding capabilities of LLMs. In these settings, autonomous agents like OpenHands (Wang et al., 2025b) and SWE-Agent (Yang et al., 2024) are tasked with exploring repositories, localizing issues, generating patches (Pull Requests), and executing tests to validate solutions. This execution-based verification is pivotal, not only for evaluation but also for emerging training paradigms like Reinforcement Learning with Verifiable Rewards (RLVR) (Wen et al., 2025). However, the efficacy of such methods is constrained by the scalability of executable environment construction. Consequently, existing efforts face a dilemma: approaches based on static code metrics (Xie et al., 2025; Wei et al., 2025) scale efficiently but provide only approximate verification signals, while manual construction (Pan et al., 2025) ensures quality but remains labor-intensive and largely restricted to Python. This leaves a critical gap for scalable, verifiable support across diverse programming languages.

---

[*]Equal contribution   [1]Research Center for Social Computing and Interactive Robotics, Harbin Institute of Technology, Harbin, China [2]Baidu Inc., Shenzhen, China. Correspondence to: Siqi Bao <baosiqi@baidu.com>, Wanxiang Che <car@ir.hit.edu.cn>.

*Proceedings of the $43^{rd}$ International Conference on Machine Learning*, Seoul, South Korea. PMLR 306, 2026. Copyright 2026 by the author(s).

To bridge this gap, we introduce **MEnvAgent**, an automated framework engineered for scalable, polyglot environment construction (see Figure 1). Our approach aims to address two fundamental challenges in this field: (1) Complexity. Managing diverse dependencies across non-standard repositories requires deep expertise. Frequent construction failures (e.g., version conflicts, compilation errors) and inconsistent testing protocols (e.g., `pytest` or `mvn test`) often lead to low success rates. (2) Time Consumption. The build process is inherently slow due to installation and compilation steps. Furthermore, environments are fragile; a single error often necessitates a costly "clean-slate" restart, creating a prohibitive overhead for large-scale data expansion.

To tackle the complexity, we design a multi-agent architecture featuring an iterative **Planning-Execution-Verification** closed loop. Within this loop, specialized agents fulfill distinct responsibilities to iteratively diagnose and autonomously resolve construction failures to ensure high success rates. To address the time consumption, we propose a novel **Environment Reuse Mechanism**. Instead of building every instance from scratch, this mechanism retrieves similar historical environments and adapts them to the target repository snapshot by synthesizing and executing incremental environment patches. This approach avoids the heavy cost of full rebuilds, thereby boosting efficiency.

Current environment construction benchmarks (Milliken et al., 2025; Eliseeva et al., 2025; Guo et al., 2025) are limited by narrow language coverage, non-executable evaluation, or insufficient quality assurance. To address these limitations and rigorously evaluate our approach, we construct **MEnvBench**, a comprehensive benchmark comprising 1,000 tasks across 10 languages, with strict execution-based evaluation and quality assurance. Extensive evaluations demonstrate that MEnvAgent outperforms state-of-the-art baselines across all languages, improving Fail-to-Pass (F2P) rates by **8.6%** while reducing time costs by **43%**. Furthermore, we leverage MEnvAgent to scale up verifiable data construction, yielding **MEnvData-SWE**, a realistic verifiable SWE training dataset. By fine-tuning open-source models on solution trajectories synthesized from this dataset, we achieve substantial performance gains on downstream SWE tasks, effectively validating the utility of MEnvAgent.

The main contributions of this paper are as follows:

- We introduce **MEnvAgent**, a multi-agent environment construction framework covering 10 programming languages, based on a Planning-Execution-Verification architecture. Notably, it incorporates a novel environment reuse mechanism that significantly reduces computational overhead.

- We present **MEnvBench**, the first comprehensive benchmark for evaluating multi-language executable environment construction. This benchmark covers 10 mainstream languages across 200 open-source repositories, comprising a total of 1,000 tasks.

- We release **MEnvData-SWE**, the largest open-source polyglot dataset of realistic verifiable Docker environments to date (see Table 13). Constructed via MEnvAgent, this dataset enables consistent performance gains for LLMs on downstream SWE tasks.

## 2. Problem Formulation

In this section, we define the task of environment construction for verifiable SWE datasets. A verifiable task instance consists of two core components: Task Context and Executable Environment. The task context is gathered from GitHub, comprising the repository snapshot $R$ and the issue with the related pull-request (PR). From the PR, we extract two distinct code changes: the *fix patch* representing logic modifications to resolve the issue, and the *test patch* containing new test cases to verify the fix. Let $R_{fix}$ denote the repository state after applying the *fix patch* to $R$.

Given a task context, the objective of environment construction is to determine a configuration triplet $(B, \mathcal{P}, T)$. Here, $B$ denotes the **base image**, $\mathcal{P}$ represents the **build process** consisting of a sequence of installation commands, and $T$ specifies the **test configuration**, which involves applying the *test patch* and executing the test command. The constructed environment is formally defined as $S = \delta(B, \mathcal{P})$, where $\delta$ represents the transition function.

The fundamental goal is executability. Specifically, the constructed environment must allow repository state $R_{fix}$ after applying the fix patch to pass the tests (**PASS**):

$$\varepsilon(R_{fix}, S, T) = 0 \tag{1}$$

Here, $\varepsilon(\cdot) = 0$ denotes that the tests are successfully passed. However, executability alone is insufficient. To ensure its validity as a verifiable environment, we enforce the Fail-to-Pass (**F2P**) criterion:

$$\varepsilon(R, S, T) = 1 \quad \wedge \quad \varepsilon(R_{fix}, S, T) = 0 \tag{2}$$

This differential outcome guarantees that the environment accurately reproduces the specific issue (Fail) and verifies its resolution (Pass). (See Appendix B for further details).

## 3. MEnvAgent Design

In this section, we introduce the design of MEnvAgent, as illustrated in Figure 2. We elaborate on two key components: the multi-agent architecture designed to construct executable environments and resolve construction failures, and the Environment Reuse Mechanism developed to accelerate this process by adapting historical environments (see Appendix C for specific details).

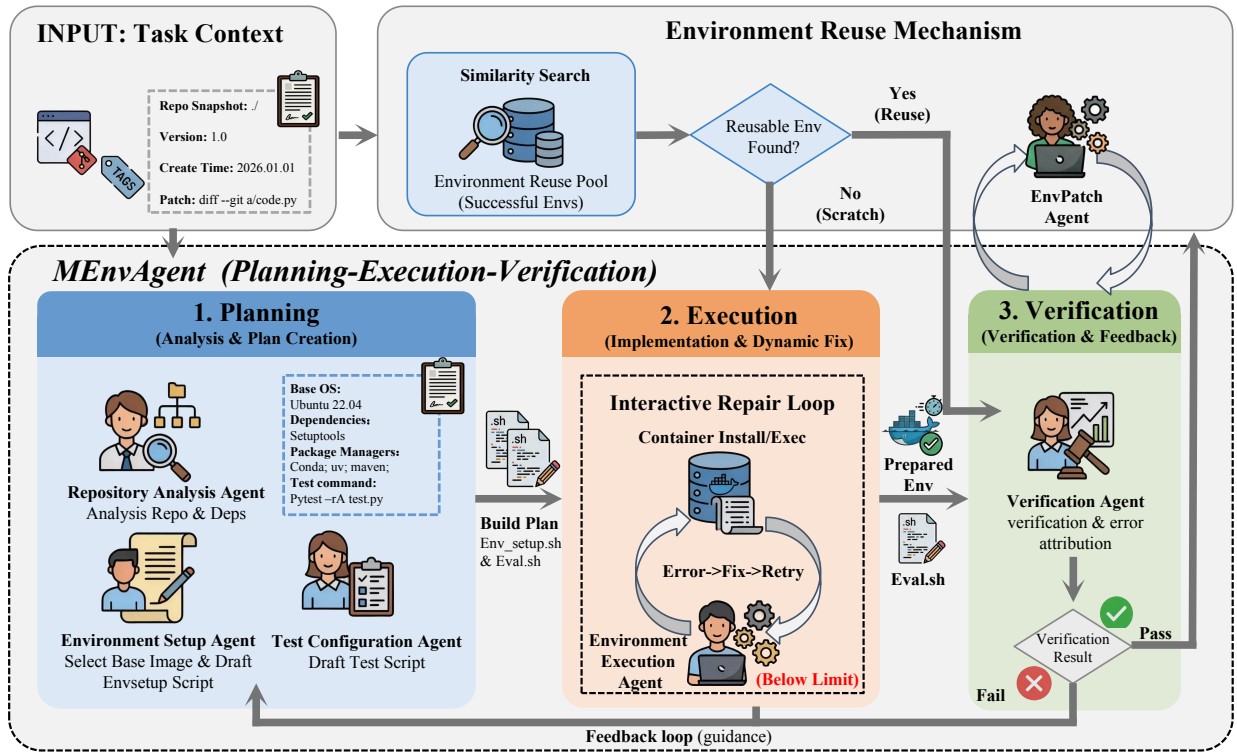

*Figure 2.* **Overview of MEnvAgent.** (Top) The Environment Reuse Mechanism retrieves and adapts historical environments via incremental patching to reduce overhead. (Bottom) The Planning-Execution-Verification loop, where agents autonomously draft scripts, interactively repair build errors, and diagnose test failures to guide iterative refinement.

### 3.1. Multi-Agent Architecture

The architecture of MEnvAgent is structured into three iterative stages: Planning, Execution, and Verification.

**Planning Stage.** This stage is orchestrated by three specialized agents to formulate the blueprint for the environment. First, the **Repository Analysis Agent** explores the file structure and contents of the target repository, producing a comprehensive summary of its project type, dependency requirements, and entry points. This summary is passed to the downstream agents. Next, the **Environment Setup Agent** determines the most suitable base image and generates a complete environment installation script (denoted as building process $\mathcal{P}$), which includes all necessary installation commands. Subsequently, the **Test Configuration Agent** analyzes the repository structure alongside the proposed installation script to synthesize a compatible test configuration script (denoted as $T$), ensuring that the verification logic aligns with the environment setup.

**Execution Stage.** Once the plan is established, the system transitions to execution. The **Environment Execution Agent** instantiates a container based on the selected image and executes the commands in $\mathcal{P}$. Crucially, this agent monitors the terminal output in real-time, capable of dynamically adjusting commands to resolve immediate execution

errors (e.g., missing packages or version conflicts). If the installation completes successfully, the workflow proceeds to the Verification Stage. However, if the agent fails to resolve installation errors after multiple attempts, the process aborts the current attempt and reverts to the Planning Stage to regenerate a new build strategy.

**Verification Stage.** The final stage verifies the correctness of the built environment $S$ and the test configuration $T$. The **Verification Agent** executes the tests defined in $T$ within the container. If the tests pass (satisfying $\varepsilon(R_{fix}, S, T) = 0$), the task is considered successful. If validation fails, the agent performs *error attribution* to diagnose whether the failure stems from a missing environment dependency or an incorrect test command. This diagnostic feedback is propagated back to the Planning Stage to guide the agents in the next iteration. Finally, we verify the successful environment against the F2P criterion (Eq. 2) to confirm its validity as a verifiable SWE environment.

### 3.2. Environment Reuse Mechanism

To reduce the computational overhead of deriving and executing the complete build process $\mathcal{P}$ from a raw base image $B$, we introduce an **Environment Reuse Mechanism**. We reformulate the problem as first identifying a similar existing

environment state $S_{sim}$ from a pool $\mathcal{S}_{pool}$ that minimizes the expected *adaptation effort* $\mathcal{C}_{adapt}$:

$$S_{sim} = \underset{S \in \mathcal{S}_{pool}}{\arg\min} \mathcal{C}_{adapt}(S, R) \tag{3}$$

Once $S_{sim}$ is retrieved, we employ an **EnvPatchAgent** to generate an incremental command sequence $\Delta\mathcal{P}$ that adapts this environment to the target repository snapshot $R$. Formally, we seek an environment patch $\Delta\mathcal{P} =$ EnvPatchAgent$(R, S_{sim})$ that transitions the retrieved environment to a valid state $S_{new}$ satisfying:

$$\varepsilon(R, S_{new}, T) = 1 \quad \wedge \quad \varepsilon(R_{fix}, S_{new}, T) = 0, \\ \text{where } S_{new} = \delta(S_{sim}, \Delta\mathcal{P}) \tag{4}$$

Our approach executes this mechanism through two stages: Environment Retrieval and Verification-Driven Adaptation.

**Environment Retrieval.** We maintain an Environment Pool $\mathcal{S}_{pool}$ containing previously verified environments. To approximate the optimal $S_{sim}$ in Eq. 3, we employ a hierarchical retrieval strategy grounded in software evolution patterns. First, we construct a candidate set based on *Version Consistency*, prioritizing historical environments associated with the exact version of the target repository snapshot. If no exact match is found, we broaden the scope to include all historical environments belonging to the same repository. Subsequently, we leverage *Backward Compatibility*, premised on the observation that newer environments typically support older dependencies. Consequently, we select an environment state newer than the target repository snapshot yet temporally closest to minimize compatibility risks.

**Verification-Driven Adaptation.** Once $S_{sim}$ is retrieved, the **EnvPatchAgent** operates within a feedback loop to generate $\Delta\mathcal{P}$. The process commences with the Test Configuration Agent synthesizing the test script $T$, which is subsequently executed within $S_{sim}$ by the Verification Agent. If execution succeeds, the environment is reused directly. However, verification failure triggers the EnvPatchAgent to analyze the diagnostic feedback and synthesize incremental commands $\Delta\mathcal{P}$. This iterative process continuously patches the environment to produce an updated state $S_{new}$, continuing until the success condition $\varepsilon(R_{fix}, S_{new}, T) = 0$ is met (see Appendix C.2 for a concrete case).

## 4. MEnvBench Construction

To address the limitations of existing benchmarks (as compared in Table 1) and rigorously evaluate our framework, we construct **MEnvBench** following a strict pipeline to ensure high quality, execution validity, and broad representativeness (see Appendix D for details).

*Table 1.* Comparison with existing benchmarks. **Langs**: Language count. **Exec-Eval**: Execution support. **Quality**: Quality Assurance. **Domain**: Domain diversity.

| Benchmark | Langs | # Repos | # Tasks | Exec-Eval | Quality | Domain |
|---|---|---|---|---|---|---|
| INSTALLAMATIC | 1 | 40 | 40 | ✓ | ✗ | ✗ |
| EXECUTIONAGENT | 5 | 50 | 50 | ✓ | ✗ | ✗ |
| EnvBench | 3 | 994 | 994 | ✗ | ✗ | ✗ |
| Repo2Run-bench | 2 | 420 | 420 | ✗ | ✗ | ✗ |
| SweSetupBench-lite | 4 | 12 | 671 | ✓ | ✗ | ✗ |
| **MEnvBench (Ours)** | **10** | **200** | **1000** | ✓ | ✓ | ✓ |

### 4.1. Data Collection and Filtering

Our data acquisition pipeline consists of two phases, transforming raw GitHub data into a high-quality candidate pool.

**Phase 1: Repository Acquisition.** We targeted high-quality repositories across 10 mainstream programming languages. To minimize construction failures stemming from inherent code defects, we applied strict criteria: repositories must have (1) $> 1,000$ stars, (2) $> 200$ forks, issues, PRs, and (3) a primary language ratio of $> 60\%$. This stage yielded a candidate pool of **8,000** repositories.

**Phase 2: Instance Extraction & Quality Assurance.** From these repositories, we extracted Issue-PR pairs spanning 2018–2025. We enforced strict quality controls, including retaining only *closed* issues explicitly linked to a PR containing a *test patch* and employing an LLM-based assessment to filter out low-quality issues (score $< 5$). This process yielded a refined pool of **213,766** instances.

### 4.2. Benchmark Composition

From the filtered candidate pool, we employed a sampling strategy to construct MEnvBench, comprising **1,000** tasks (10 languages $\times$ 20 repositories $\times$ 5 instances selected from distinct historical versions). This allocation structure strikes a strategic balance between *inter-project breadth* and *intra-project depth*: it ensures coverage of diverse repository types while capturing sufficient internal variability to verify build robustness. To ensure comprehensive representativeness, we selected repositories based on two key dimensions:

- **Domain Diversity:** We leveraged LLMs to classify repositories into specific domains (e.g., AI, System). Our sampling prioritizes wide coverage to ensure robustness across diverse software ecosystems (see Figure 10a).
- **Project Scale:** We sampled across five size bands (from $<$10MB to $>$500MB) to encompass a full spectrum of difficulty levels, as repository size typically correlates with build complexity (see Figure 10b).

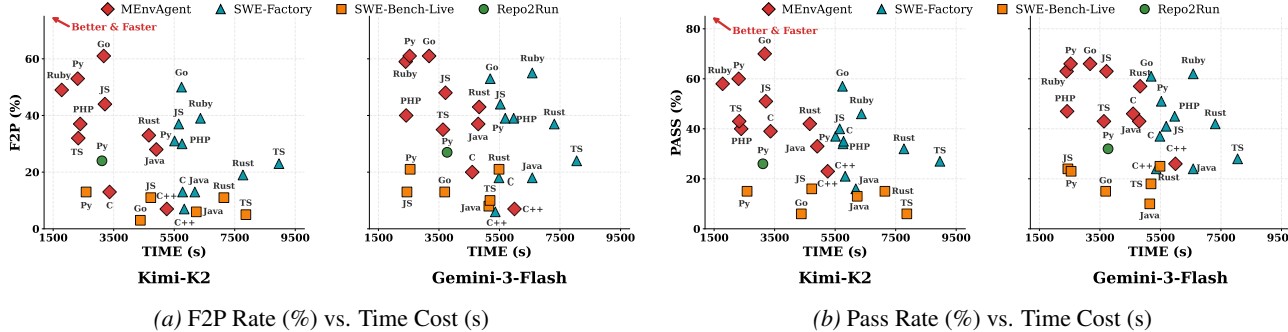

*(a)* F2P Rate (%) vs. Time Cost (s)          *(b)* Pass Rate (%) vs. Time Cost (s)

*Figure 3.* **Performance trade-off analysis on MEnvBench.** The x-axis represents the average time cost (lower is better), and the y-axis represents the pass rate (higher is better). MEnvAgent points cluster in the top-left region, indicating it achieves higher validity and success rates with significantly lower time consumption compared to baselines.

### 4.3. Evaluation Metrics

We employ three metrics to evaluate the performance of our framework:

- **Pass Rate (PASS):** The percentage of tasks satisfying the **Executability** condition (Eq. 1).
- **Fail-to-Pass Rate (F2P):** The percentage of tasks satisfying the strict **Validity** criterion (Eq. 2).
- **Time Cost (TIME):** The average wall-clock time consumed per task, reflecting the efficiency of the environment construction process.

## 5. Experiment

We design our experiments to answer the primary research question: How does MEnvAgent compare to state-of-the-art baselines in terms of environment construction success rates and computational efficiency across diverse programming languages on MEnvBench?

**Model Details.** To assess the robustness and generalization of our framework, we employ two representative Large Language Models (LLMs) as the reasoning backbone for the agents. For the open-source model, we select Kimi-K2 (kimi-k2-0905-preview) (Team et al., 2025), which demonstrates superior capability in agentic planning and long-context understanding. For the closed-source model, we utilize Gemini-3-Flash (Google, 2025). We selected this model because it represents the latest state-of-the-art capabilities while maintaining low latency and high cost-efficiency, which are critical prerequisites for scalable environment construction scenarios.

**Baseline Methods.** We compare MEnvAgent against three categories of baselines: (1) **Repo2Run** (Hu et al., 2025), a Python-specialized tool evaluated exclusively on the Python subset due to its extensibility constraints; (2) **SWE-Bench-Live** (Zhang et al., 2025a), which supports 6 of the languages in MEnvBench, allowing for a multi-

*Table 2.* Averaged performance comparison across 10 languages.

| Method | MEnvBench | | |
|---|---|---|---|
| | **F2P (%)** | **PASS (%)** | **TIME (s)** |
| *Kimi-K2* | | | |
| SWE-Factory | 26.2 | 34.5 | 6356 |
| **MEnvAgent** | **35.7** (+9.5) | **45.9** (+11.4) | **3339** (-47.5%) |
| *Gemini-3-Flash* | | | |
| SWE-Factory | 33.3 | 41.5 | 6175 |
| **MEnvAgent** | **41.1** (+7.8) | **52.0** (+10.5) | **3808** (-38.3%) |
| *Average* | | | |
| SWE-Factory | 29.8 | 38.0 | 6266 |
| **MEnvAgent** | **38.4** (+8.6) | **49.0** (+11.0) | **3574** (-43.0%) |

language sub-evaluation; and (3) **SWE-Factory** (Guo et al., 2025), a state-of-the-art agent framework. We evaluated this baseline across all 10 languages in MEnvBench. For detailed hyperparameter settings, please refer to Appendix E.

**Results on MEnvBench.** We evaluate the overall effectiveness and efficiency of our framework on MEnvBench. The aggregated metrics are presented in Table 2, and the efficiency-quality trade-off is visualized in Figure 3. In the scatter plots, the x-axis represents the average time cost, while the y-axis denotes the Pass Rate or F2P Rate. As illustrated, MEnvAgent consistently occupies the upper-left quadrant across all backbone models and programming languages. This distribution signifies an optimal performance state — simultaneously achieving the highest validity while minimizing computational overhead. In contrast, the baselines exhibit distinct limitations: SWE-Factory is predominantly distributed in the right-hand region, indicating that while it achieves competitive runnability, it suffers from excessive latency due to inefficient trial-and-error loops. Meanwhile, Repo2Run and SWE-Bench-Live cluster in the lower region, where despite maintaining acceptable efficiency, their capability to generate valid environments is significantly compromised. This visual superiority is

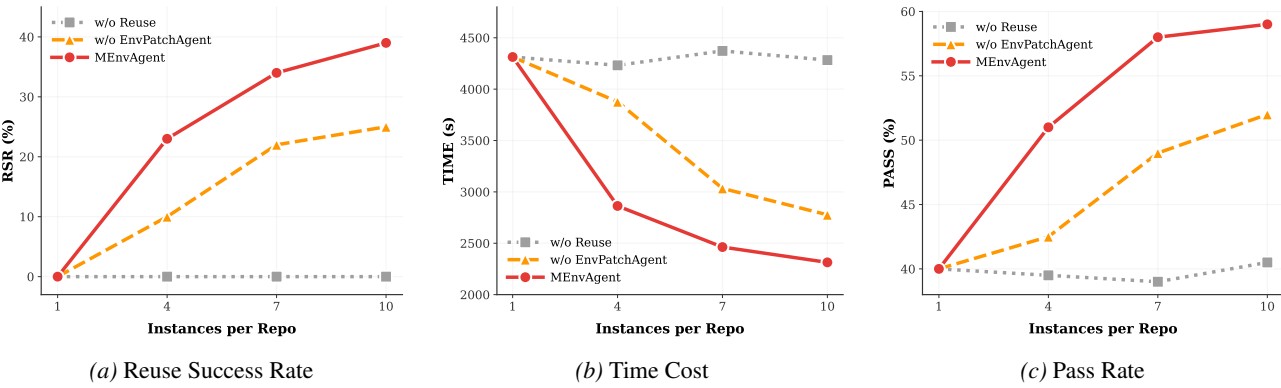

*(a)* Reuse Success Rate       *(b)* Time Cost       *(c)* Pass Rate

*Figure 4.* **Impact of data scale on performance metrics.** We illustrate the trends of (a) Reuse Success Rate, (b) Time Cost, and (c) Pass Rate as the number of instances per repository increases from 1 to 10. The results confirm that larger data scale significantly enhances reuse probability and overall efficiency.

quantitatively corroborated by Table 2, which compares our method directly against the strongest baseline, SWE-Factory. Averaged across models, MEnvAgent improves the strict F2P Rate by **8.6%** and the Pass Rate by **11.0%**, while simultaneously reducing time costs by **43.0%**. For complete per-language statistics and granular comparisons, we refer readers to Table 9 in Appendix F.

## 6. Analysis

### 6.1. Ablation Study on Environment Reuse

To validate the effectiveness of the Environment Reuse mechanism and the critical role of the EnvPatchAgent, we conduct a comprehensive ablation study. Furthermore, we investigate how the data scale (the number of historical instances per repository) influences reuse performance.

**Experimental Setup.** To isolate the contribution of each component, we evaluate our framework against two ablated variants: (1) **MEnvAgent (Full)**, our complete framework incorporating both the Environment Retrieval and the EnvPatchAgent; (2) **w/o EnvPatchAgent (Direct)**, a variant that retrieves the most similar environment $S_{sim}$ but applies it directly without modification, validating the necessity of the patching mechanism; and (3) **w/o Reuse (Scratch)**, the minimal baseline that disables the reuse mechanism entirely and builds every task from the base image. To measure reuse efficacy, we employ the **Reuse Success Rate (RSR)**, defined as the proportion of tasks successfully verified via the reuse pathway without falling back to the scratch build. All ablation experiments are conducted using Kimi-K2 on a Python subset from MEnvBench, with the data scale extended to 10 instances per repository.

**Component Effectiveness.** We analyze the contribution of each component in Table 3. MEnvAgent achieves optimal performance across all metrics, validating the syn-

*Table 3.* **Ablation results on component effectiveness** (10 instances per repository). **RSR** denotes the Reuse Success Rate.

| Method | RSR (%) | PASS (%) | TIME (s) |
|---|---|---|---|
| **MEnvAgent** | **39.0** | **59.0** | **2314** |
| w/o EnvPatchAgent | 25.0 | 52.0 | 2777 |
| w/o Reuse | 0.0 | 40.5 | 4283 |

ergy between retrieval and patching. In terms of efficiency and reuse stability, removing the EnvPatchAgent (*w/o EnvPatchAgent*) causes the Reuse Success Rate to drop significantly from 39.0% to 25.0%, leading to a 20% increase in time cost due to frequent fallbacks to scratch builds. Furthermore, compared to the baseline without reuse (*w/o Reuse*), our full framework reduces the average computational time by 46.0%. Crucially, beyond these efficiency gains, MEnvAgent significantly boosts the overall Pass Rate by 18.5% compared to the *w/o Reuse* baseline. This improvement stems from the reuse mechanism, which avoids the error-prone process of resolving complex dependencies from scratch (see Appendix G for the comprehensive multilingual ablation results).

**Impact of Data Scale.** We further investigate how the volume of historical data influences performance by scaling the number of instances per repository from 1 to 10, as shown in Figure 4. In sparse data settings (1 instance), the Reuse Success Rate is negligible, resulting in performance similar to the scratch baseline. However, as the data scale expands to 10 instances, the Reuse Success Rate rises steadily to 39%, driving concurrent improvements in Pass Rate and corresponding reductions in Time Cost. This trend highlights the scalability of our approach, suggesting that in real-world scenarios characterized by large-scale data accumulation, the framework is poised to deliver even greater efficiency gains.

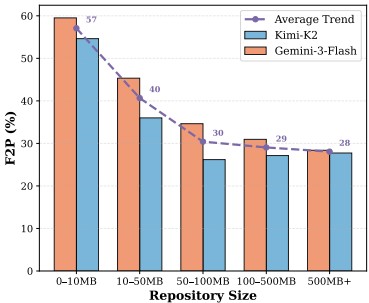

*Figure 5.* F2P performance analysis relative to repository size.

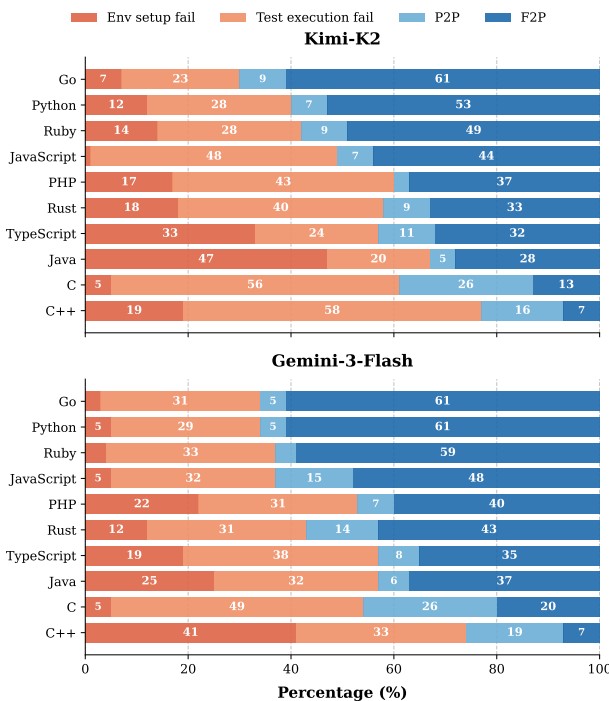

*Figure 6.* Error distribution across 10 programming languages.

## 6.2. In-depth Result Analysis

**Performance vs. Repository Scale.** Figure 5 analyzes the correlation between model performance and repository characteristics. We observe a significant negative correlation between Fail-to-Pass (F2P) rates and repository size. This trend is attributable to the intricate dependency graphs and substantial build overheads inherent in large-scale projects, which exacerbate the complexity of automated environment configuration.

**Error Distribution and Behavioral Patterns.** Figure 6 categorizes task outcomes for Kimi-K2 and Gemini-3-Flash into four states: Fail-to-Pass (F2P), Pass-to-Pass (P2P), Test Execution Failure, and Environment Setup Failure. Our analysis reveals a significant cross-language performance disparity. Modern languages with standardized package ecosystems, such as Go and Python, demonstrate high resolution rates (F2P), indicating that current LLMs effectively handle dependency management. In contrast, discrepancies emerge in complex ecosystems like Java. Gemini-3-Flash exhibits superior robustness in environment setup, consistently maintaining lower setup failure rates compared to Kimi-K2 across most languages. This advantage is most pronounced in Java, where Gemini reduces the setup failure rate by nearly half relative to Kimi, suggesting better generalization in generating intricate build scripts (e.g., Maven/-Gradle configurations). Conversely, C-family languages (C/C++) are dominated by compilation errors derived from complex CMake configurations and high resource consumption, which frequently lead to timeouts (see Appendix H for a detailed failure analysis). These diverse failure patterns underscore the necessity of the Verification Agent within the MEnvAgent framework to enable precise error attribution and iterative refinement beyond initial setup.

## 6.3. Scaling Verifiable SWE Datasets via MEnvAgent

To validate the utility of **MEnvAgent** for training software engineering agents, we employ rejection sampling fine-tuning as the primary procedure for improving base LLMs, following the methodology established in prior works (Yang

et al., 2025c; Guo et al., 2025; Wang et al., 2025a). Our experiment workflow is as follows: First, we leverage MEnvAgent to establish a fully automated pipeline to **scale up** the construction of verifiable software engineering tasks from real-world GitHub repositories. Through this pipeline, we construct **MEnvData-SWE**, a diverse dataset comprising 3,005 task instances from 942 repositories across 10 programming languages, all equipped with executable environments. Next, we deploy an agent framework with an expert model on MEnvData-SWE to collect solution trajectories. Finally, we fine-tune the student model on 3,872 trajectories derived from resolved instances and evaluate them on separate benchmarks (see Appendix I for details).

**Models.** For the expert model, we employ Claude-4.5-Sonnet (Anthropic, 2025), representing the state-of-the-art in coding capabilities. For student models, we select a diverse array of architectures to verify robustness: the dense Qwen-2.5-Coder-Instruct series (7B, 14B, and 32B) (Hui et al., 2024), the MoE-based Qwen-3-Coder-30B-A3B-Instruct, and GLM-4.5-Air (Zeng et al., 2025). Training details are provided in Appendix I.6.

**Agent Scaffolding.** We use **OpenHands** (Wang et al., 2025b), an event-driven framework that provides a sandboxed environment where agents interact with the codebase by editing files (`str-replace-editor`), executing shell commands (via `execute-bash`), or submitting the task (via `finish`). We selected OpenHands as it has es-

*Table 4.* **Performance on SWE-bench Verified and SWE-bench Multilingual.** Performance is measured by Resolved Rate (%).

| Category | Model | SWE-bench Verified | | SWE-bench Multilingual | |
|---|---|---|---|---|---|
| | | Baseline | SFT | Baseline | SFT |
| **Reference Models** | GPT-4.1 | 54.6 | - | 31.5 | - |
| | Claude-4.5-Sonnet | 77.2 | - | 68.0 | - |
| **Our Fine-tuned Models** | Qwen2.5-Coder-7B-Instruct | 0.0 | **21.8** (+21.8) | 0.0 | **12.3** (+12.3) |
| | Qwen2.5-Coder-14B-Instruct | 5.8 | **39.8** (+34.0) | 0.0 | **31.2** (+31.2) |
| | Qwen2.5-Coder-32B-Instruct | 7.5 | **54.6** (+47.1) | 0.0 | **38.3** (+38.3) |
| | Qwen3-Coder-30B-A3B-Instruct | 45.2 | **53.4** (+8.2) | 34.7 | **38.0** (+3.3) |
| | GLM-4.5-Air | 58.0 | **62.8** (+4.8) | 42.3 | **47.7** (+5.4) |

tablished strong baselines on benchmarks like SWE-bench.

**Evaluation Benchmarks.** We evaluate on SWE-bench Verified (Chowdhury et al., 2024) (500 curated Python tasks) and SWE-bench Multilingual (Yang et al., 2025c) (encompassing 9 languages), reporting the Resolved Rate (%). It is worth noting that we rectified a `git log` manipulation issue to prevent potential data leakage[1]. Consequently, our reported scores may be slightly lower than the official baselines due to this more rigorous evaluation setting.

**Results.** Table 4 demonstrates that fine-tuning on our dataset consistently boosts performance across all models. Within the Qwen2.5-Coder series, we observe a positive correlation between model scale and performance gains. Strikingly, Qwen2.5-Coder-32B matches GPT-4.1 (OpenAI, 2025) on SWE-bench Verified and significantly outperforms it on the Multilingual benchmark. Furthermore, we achieve substantial gains even on MoE baselines (Qwen3-Coder and GLM-4.5-Air) that have already been heavily optimized with agentic data. This confirms that scaling verifiable data effectively pushes the performance boundaries of SOTA models, strongly validating the utility of MEnvAgent.

## 7. Related Work

**Automated Environment Construction.** Early attempts at automated environment setup primarily relied on static heuristics to infer dependencies from source code, offering determinism but struggling with complex configurations and version incompatibilities (Gruber & Fraser, 2023; Zhang et al., 2024; Yang et al., 2025b). With the rapid evolution of Large Language Models (LLMs), a series of LLM-based automated approaches have emerged to address these limitations (Bouzenia & Pradel, 2025; Milliken et al., 2025; Vergopoulos et al., 2025; Badertdinov et al., 2025). Among the works most relevant to ours, Repo2Run (Hu et al., 2025) employs a dual-agent framework tailored with Python-specific tools, focusing exclusively on environment installation via

___
[1]https://github.com/SWE-bench/SWE-bench/issues/465

fixed test commands that do not execute verification tests. Similarly, SWE-Bench-Live (Zhang et al., 2025a) extends the task scope to encompass both environment setup and test configuration, utilizing a single-agent method via interactive bash sessions. In contrast, SWE-Factory (Guo et al., 2025) further broadens applicability by supporting four programming languages, introducing a collaborative multi-agent architecture for automated environment construction.

**Environment Construction Benchmarks.** Initial efforts evaluated environment construction capability implicitly within comprehensive tasks (Bogin et al., 2024; Siegel et al., 2024; Tang et al., 2025), offering little insight into the isolated challenges LLMs face during the construction phase. To explicitly assess this capability, dedicated benchmarks emerged: INSTALLAMATIC (Milliken et al., 2025) and EXECUTIONAGENT (Bouzenia & Pradel, 2025) established rigorous execution-based standards but remained small-scale. Conversely, EnvBench (Eliseeva et al., 2025) and Repo2Run-bench (Hu et al., 2025) scaled up data but relied on approximate evaluation metrics like static compilation checks or test collection, which often fail to detect runtime incompatibilities essential for robust agent feedback. To further enhance diagnostic depth, EnConda-Bench (Kuang et al., 2025) introduces process-level diagnostics, yet it remains restricted to Python and rigid configuration patterns. Notably, SweSetupBench-lite (Guo et al., 2025) aligns evaluation with realistic software evolution by supporting historical repository snapshots and adopting metrics like Fail-to-Pass (F2P) rates and efficiency costs. While these features suit scalable evaluation scenarios and capture dependency drift, the representativeness of the benchmark is hindered by a limited scope of just 12 repositories.

## 8. Conclusion

In this paper, we introduced MEnvAgent, a polyglot framework that automates the complex task of environment construction through a multi-agent Planning-Execution-Verification architecture. Notably, it incorporates a novel

environment reuse mechanism that significantly reduces computational overhead. We rigorously evaluated our framework on MEnvBench, a new benchmark comprising 1,000 tasks across 10 programming languages, where MEnvAgent demonstrated superior performance, achieving significantly higher F2P rates and lower time costs compared to state-of-the-art baselines. Furthermore, to validate the utility of our approach, we leveraged MEnvAgent to construct MEnvData-SWE. Experiments demonstrate that models fine-tuned on this dataset achieve substantial performance gains on SWE tasks. Finally, we open-source our code, benchmark, and dataset to facilitate future research in the community.

## Impact Statement

We expect **MEnvAgent** to offer significant advantages by automating the traditionally labor-intensive task of environment construction, thereby lowering the barrier for researchers to conduct large-scale, polyglot software engineering studies. By scaling up execution-verified data construction across 10 programming languages, our framework empowers the community to develop more robust and versatile coding agents. However, we acknowledge the potential risks associated with automated environment setup and code execution. Malicious actors could potentially exploit the framework to construct environments for developing harmful software or executing unauthorized scripts. Furthermore, LLMs may inevitably generate erroneous commands that could lead to severe consequences. To address this inherent risk, our framework executes all tasks within a strictly isolated Docker sandbox environment by default. We strongly recommend that users adhere to this default configuration to ensure system safety and isolation. Finally, all datasets and models utilized in this study are open-source and adhere strictly to their respective licenses. We hope our findings and contributions will catalyze future research in this field and foster the responsible advancement of AI technologies within the software engineering domain.

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

# Appendix

# A. Extended Related Work

In this section, we provide a more granular discussion on the landscape of verifiable software engineering, focusing specifically on the evolution of execution-based benchmarks and the emerging domain of verifiable training dataset construction.

## A.1. Verifiable SWE Benchmarks

The foundational SWE-bench (Jimenez et al., 2024) established the standard for execution-based evaluation, focusing on issue resolution within Python repositories. Recognizing the need for broader language support, subsequent works such as SWE-bench Multilingual (Yang et al., 2025c) and Multi-SWE-bench (Zan et al., 2025) extended this paradigm to polyglot environments, incorporating languages like Java, JavaScript, and Go. Beyond textual code changes, SWE-bench Multimodal (Yang et al., 2025a) introduced visual debugging tasks, adding a new dimension to agent evaluation. Furthermore, the scope of tasks has diversified beyond bug fixing: FEA-Bench (Li et al., 2025) targets feature implementation, while SWE-Perf (He et al., 2025) focuses on code optimization and performance enhancement.

**Connection to MEnvAgent:** The maintenance and expansion of these benchmarks heavily rely on successful environment construction. MEnvAgent can significantly accelerate this process, enabling the continuous update of these benchmarks with fresh, real-world repositories to prevent data contamination and stagnation.

## A.2. Verifiable SWE Training Datasets

To improve agent performance, recent research has pivoted towards constructing verifiable training datasets. SWE-gym (Pan et al., 2025) demonstrates the value of rigorous verification but relies on manual curation, limiting its scale. To overcome scalability bottlenecks, SWE-Smith (Yang et al., 2025c) utilizes a limited set of base environments and injects synthetic bugs via code mutation to mass-produce tasks. In a different approach, SWE-Flow (Zhang et al., 2025b) introduces a synthesis framework grounded in Test-Driven Development (TDD); instead of relying on human-submitted issues, it automatically infers incremental development steps directly from unit tests. Similarly, SWE-Synth (Pham et al., 2025) proposes an agent-based framework that simulates human debugging workflows to synthesize human-like bugs at the repository level. More recently, SWE-mirror (Wang et al., 2025a) addresses the "sim-to-real" gap by reproducing real-world GitHub issues within containerized environments, ensuring data authenticity.

**Connection to MEnvAgent:** Unlike approaches that rely on synthetic mutations or limited base environments, MEnvAgent enables the scaling of training data directly from diverse, real-world scenarios. Our work is orthogonal to methods like SWE-Smith, SWE-Flow and SWE-Flow; by providing a massive pool of successfully built environments, MEnvAgent can serve as the foundational infrastructure to further boost their generation pipelines.

# B. Detailed Problem Formulation

In this appendix, we provide the formal mathematical definitions for the task of executable environment building. Building upon the formulation in Repo2Run (Hu et al., 2025), we extend the notation to support the **joint synthesis** of the build process and test configuration.

## B.1. State Transition Dynamics

**Environment State** ($S$). Let $\mathcal{S}$ denote the set of all possible environment states. An environment state $S \in \mathcal{S}$ represents a comprehensive snapshot of the computer system, encompassing all variables, files, installed packages, and system caches.

**Command Sequence** ($C$). Let $\mathcal{C}$ denote the set of all possible command sequences. A command sequence $C \in \mathcal{C}$ consists of a series of individual instructions (e.g., shell commands) that, when executed, modify the system state.

**State Transition Function** ($\delta$). We define the state transition as a deterministic function $\delta : \mathcal{S} \times \mathcal{C} \to \mathcal{S}$. It maps a starting state $S_{start}$ and a command sequence $C$ to a resulting state $S_{end}$:

$$\delta(S_{start}, C) = S_{end} \tag{5}$$

This function encapsulates the execution of commands (e.g., via a bash interface) that transform the system environment.

### B.2. Base Image Initialization

**Empty State ($S_\emptyset$).** The empty state $S_\emptyset \in \mathcal{S}$ represents a bare-metal operating system or a hypothetical null state with no user-level configurations.

**Base Image ($B$).** A base image $B \in \mathcal{S}$ is a specific environment state, typically pre-configured for convenience (e.g., `python:3.10`). Formally, a base image $B$ is reachable from the empty state $S_\emptyset$ via a predefined command sequence $C_B \in \mathcal{C}$:

$$B = \delta(S_\emptyset, C_B) \tag{6}$$

In our framework, the selection of $B$ is the first step in the construction pipeline.

### B.3. Building Process and Verification

**Building Process ($\mathcal{P}$).** The building process $\mathcal{P} \in \mathcal{C}$ is a synthesized command sequence designed to install dependencies and configure the environment starting from the base image $B$. The final environment state $S$ is obtained by:

$$S = \delta(B, \mathcal{P}) \tag{7}$$

**Test Configuration ($T$).** Unlike prior works that assume fixed test commands, we define $T$ as a synthesized specification that includes the test entry points and execution arguments tailored to the repository.

**State Verification ($\varepsilon$).** The verification function $\varepsilon$ determines whether the constructed environment $S$ is valid for a given repository $R$ under the test configuration $T$. We define $\varepsilon$ as a Boolean function:

$$\varepsilon(R, S, T) = \begin{cases} 0 & \text{if all tests defined in } T \text{ pass in state } S \\ 1 & \text{otherwise} \end{cases} \tag{8}$$

Therefore, the goal of our task is to find the triplet $(B, \mathcal{P}, T)$ such that $\varepsilon(R, \delta(B, \mathcal{P}), T) = 0$.

## C. MEnvAgent Implementation Details

This appendix details the technical implementation of MEnvAgent, including agent specifications, the core algorithm, and a concrete case study.

### C.1. Agent Specifications and Workflow

In this section, we provide additional technical specifications for the **MEnvAgent** framework to ensure reproducibility and clarity of the multi-agent interactions. First, Table 5 presents the granular **Input-Output** (I/O) specifications for each specialized agent, detailing the specific information consumed and the artifacts generated during the environment construction process. Subsequently, Algorithm 1 outlines the comprehensive operational workflow of MEnvAgent. This includes the logic for the **Environment Reuse Mechanism** (Phase 1), which leverages historical environments to minimize overhead, and the **Iterative Construction Loop** (Phase 2), which coordinates the planning, execution, and verification stages to autonomously resolve complex build failures.

### C.2. Case Study: Environment Reuse Process

In this section, we present a detailed execution trace to illustrate the practical operation of the **Environment Reuse Mechanism**. The case study, detailed in Figure 7, focuses on a real-world scenario from the `home-assistant-core` repository. Here, the system attempts to reuse a verified historical environment to execute a new test case. The trace demonstrates the collaborative process between the **Verification Agent** and the **EnvPatchAgent**: the former identifies a missing dependency through execution failure, while the latter synthesizes a context-aware incremental patch ($\Delta \mathcal{P}$) by analyzing the original build logic. This process exemplifies how MEnvAgent achieves rapid environment adaptation without the prohibitive cost of rebuilding base images from scratch.

*Table 5.* Detailed Input and Output Specifications for MEnvAgent Components.

| Agent Name | Input Information | Generated Outputs |
|---|---|---|
| **Repository Analysis Agent** | **Target Repository Snapshot**($R$)**:** File tree structure, key configuration files (e.g., `package.json`, `CMakeLists.txt`), and documentation. | **Project Summary:** Extracted metadata including primary language, build system type, and identified dependencies. |
| **Environment Setup Agent** | **1. Project Summary:** From Repository Analysis Agent. 
 **2. Feedback:** Diagnosis from Verification Agent (in retry loops). | **Build Plan:** Selected Base Image ($B$) and the complete Setup Script ($\mathcal{P}$) containing installation commands. |
| **Test Configuration Agent** | **1. Project Summary:** To identify test frameworks. 
 **2. Setup Script** ($\mathcal{P}$)**:** To align test commands with installed binaries. 
 **3. Feedback:** Diagnosis from Verification Agent (in retry loops). | **Test Script** ($T$)**:** Executable commands to trigger the repository's test suite, including necessary environment variables. |
| **Environment Execution Agent** | **1. Base Image** ($B$)**:** Docker image context. 
 **2. Setup Script** ($\mathcal{P}$)**:** Commands to execute. | **Runtime Environment** ($S$)**:** A built container instance (if successful). 
 **Execution Logs:** `stdout`/`stderr` streams (if failed). |
| **Verification Agent** | **1. Environment** ($S$)**:** The built container. 
 **2. Test Script** ($T$)**:** Commands to validate correctness. | **1. Result:** Boolean success status. 
 **2. Diagnosis:** Error attribution report (e.g., "Missing Dependency") used as Feedback for planning agents. |
| **EnvPatchAgent** 
 *(Reuse Mechanism)* | **1. Target Repository** ($R$) 
 **2. Similar Env** ($S_{sim}$)**:** Retrieved historical env. 
 **3. Feedback:** From verification failure in $S_{sim}$. | **Incremental Patch** ($\Delta\mathcal{P}$)**:** A sequence of commands to adapt $S_{sim}$ to satisfy $R$'s requirements. |

# D. MEnvBench Construction Details

To ensure the high quality and reproducibility of MEnvBench, we implemented a rigorous data acquisition pipeline. This pipeline, which also serves as the foundation for the MEnvData-SWE dataset (see Appendix I), consists of three stages: Collection, Filtering, and Quality Evaluation.

## D.1. Data Collection and Filtering Strategy

We target high-quality repositories and Issue-Pull Request (PR) pairs from GitHub. To filter out noise and ensure the tasks are solvable, we apply a set of strict heuristic rules for both repositories and instances, as detailed in Table 6. Notably, we require the presence of both test patches and fix patches to enable execution-based verification.

*Table 6.* Filtering criteria for Repositories and Instances used in our pipeline.

| Scope | Metric / Criteria | Threshold |
|---|---|---|
| **Repository** | Popularity (Stars) | $\geq 1{,}000$ |
| | Primary Language Ratio | $\geq 60\%$ |
| | Community Activity (Forks, Issues, PRs) | $\geq 200$ each |
| **Instance** | Issue Status | Closed |
| | Problem Description | Non-empty |
| | Test Patch & Fix Patch | Non-empty |
| | Code Modification | Required |
| | Patch Size (Lines of Code) | $\leq 1{,}000$ |
| | Scope (Number of Files) | $\leq 10$ |

---

**Algorithm 1** MEnvAgent Environment Construction Workflow

---

**Input:** Target Repository $R$, Environment Pool $\mathcal{S}_{pool}$
**Output:** Valid Environment $S$ or Failure
 1: *// Phase 1: Environment Reuse Mechanism*
 2: $S_{sim} \leftarrow$ RETRIEVESIMILARENV$(R, \mathcal{S}_{pool})$
 3: **if** $S_{sim} \neq$ Null **then**
 4:     $T \leftarrow$ TESTCONFIGAGENT$(R)$
 5:     $Verified, Logs \leftarrow$ VERIFICATIONAGENT$(S_{sim}, T)$
 6:     **if** $Verified$ is **True then**
 7:         **return** $S_{sim}$
 8:     **else**
 9:         *// Reuse failed, attempt patching*
10:         $\Delta\mathcal{P} \leftarrow$ ENVPATCHAGENT$(R, S_{sim}, Logs)$
11:         $S_{new} \leftarrow$ EXECUTE$(S_{sim}, \Delta\mathcal{P})$
12:         **if** VERIFICATIONAGENT$(S_{new}, T)$ is **Success then**
13:             **return** $S_{new}$
14:         **end if**
15:     **end if**
16: **end if**
17:
18: *// Phase 2: Iterative Construction*
19: $Feedback \leftarrow \emptyset$
20: **for** $i \leftarrow 1$ **to** $MaxRetries$ **do**
21:     **Stage 1: Planning**
22:     $Summary \leftarrow$ REPOANALYSISAGENT$(R)$
23:     $\mathcal{P}, B \leftarrow$ ENVSETUPAGENT$(Summary, Feedback)$
24:     $T \leftarrow$ TESTCONFIGAGENT$(Summary, \mathcal{P}, Feedback)$
25:     **Stage 2: Execution**
26:     $S, Status, ExecLogs \leftarrow$ ENVEXECAGENT$(B, \mathcal{P})$
27:     **if** $Status$ is **Failure then**
28:         $Feedback \leftarrow ExecLogs$
29:         **continue**
30:     **end if**
31:     **Stage 3: Verification**
32:     $Verified, Diagnosis \leftarrow$ VERIFICATIONAGENT$(S, T)$
33:     **if** $Verified$ is **True then**
34:         $\mathcal{S}_{pool} \leftarrow \mathcal{S}_{pool} \cup \{S\}$
35:         **return** $S$
36:     **else**
37:         $Feedback \leftarrow Diagnosis$
38:     **end if**
39: **end for**
40: **return Failure**

---

## D.2. Automated Quality Evaluation

Heuristic filtering alone cannot guarantee the semantic clarity of issue descriptions. To eliminate vague or irrelevant tasks, we employ an LLM-based evaluator (DeepSeek-V3.2). As shown in the prompt template in Figure 8, the model acts as a judge, scoring the completeness of the problem description. We strictly discard any instances with a score lower than the threshold of 5.

---

**Case Study: Environment Reuse Process (home-assistant-core)**

**Phase 1: Initial Verification (Failure)**          *Agent: Verification Agent*

The agent attempts to execute the new test case in the historical environment without modification.

```
$ bash eval.sh
tests/components/rainbird/conftest.py:10:  in <module>
from pyrainbird import encryption
E   ModuleNotFoundError:  No module named 'pyrainbird'
EXIT_CODE=4
```

**Phase 2: Error Diagnosis**          *Agent: Verification Agent*

The agent identifies the missing dependency and generates specific guidance.

**Diagnosis Output (JSON):**
```
{ "error_category":  "dependency_error",
"guidance":  "Missing 'pyrainbird'.  1.  Activate conda:  source ...  && conda
activate testbed; 2.  Install:  pip install pyrainbird" }
```

**Phase 3: Patch Generation**          *Agent: EnvPatchAgent*

Based on the guidance and original setup script context (detecting Conda + pip usage), the agent generates a minimal incremental patch.

**Generated Patch ($\triangle \mathcal{P}$):**
```
#!/usr/bin/env bash
set -euxo pipefail
source /opt/miniconda3/etc/profile.d/conda.sh
conda activate testbed # Context-aware activation
pip install pyrainbird # Incremental fix
```

**Phase 4: Final Verification (Success)**          *Agent: Verification Agent*

The patch is applied, and the tests are re-executed successfully.

```
[PATCH] Installing pyrainbird package...  Successfully installed.
tests/components/rainbird/test_switch.py::test_has_unique_id PASSED
================ 11 passed in 0.65s ================
EXIT_CODE=0
```

*Figure 7.* An execution trace of the Environment Reuse Mechanism. MEnvAgent successfully adapts a historical environment by identifying a missing dependency and generating a context-aware patch (Phase 3) without rebuilding the base image.

### D.3. Data Collection Statistics

Table 7 presents the detailed statistics of our data acquisition pipeline across 10 programming languages. The "Filtered" columns represent the high-quality candidate pool (repositories with $> 1k$ stars, $> 200$ forks and PRs; instances with closed issues, PRs, and test patches) from which the final MEnvBench was sampled. The remaining high-quality instances were leveraged to construct **MEnvData-SWE**.

### D.4. Repository Diversity Analysis

To ensure MEnvBench covers a diverse range of software ecosystems, we implemented a multi-dimensional selection strategy. A key component of this strategy is **Domain Diversity**.

We leveraged Large Language Models (LLMs) to automatically classify the filtered repositories into 10 distinct domains (e.g., Machine Learning & AI, Database Systems, Web Application) based on their metadata, including repository name, description, topics, and primary language. This classification allows us to sample tasks that simulate development scenarios across various industries and technical stacks. Figure 9 illustrates the specific prompt template used for this domain classification task.

---

**Prompt for Issue Quality Evaluation**

**Role Definition**
You are an experienced software engineer. You need to evaluate the quality of an Issue to determine if it contains sufficient information for an engineer to unambiguously provide a solution.

**Issue Scoring Standards**
The full score is 10 points (Excellent quality: clear description, explicit requirements for the solution), and 0 points indicates very poor quality (impossible to understand the problem).
*Note: If a solution cannot be implemented based on the Issue, the score should not exceed 5.*

**Deduction Rules**
Check for deduction items item by item. If the issue has problems or violates the standards, point it out and deduct points in the subsequent evaluation.
**1. Major Deductions (Violating any item results in a 5-point deduction):**
- *Key Information Missing:* (1) Lack of expected results: No description of correct behavior/output; for data processing, missing input examples and expected/error outputs. (2) Lack of reproduction steps: No operation flow or runnable code to reproduce the issue. (3) Missing version info: Unspecified libraries, frameworks, OS, or environment details. (4) Incomplete error logs: Snippets provided without context or stack traces.
- *Non-Issue Type Submission:* (1) Misuse of PR description: Using Pull Request description text directly as an Issue. (2) Solved problem: The problem described is already fixed or closed. (3) Non-problem inquiry: Content involves release plans, future feature questions, etc., rather than actual defects.

**2. Common Deductions (Deduct points based on severity):**
- *Unclear Description:* (1) Mixed problems: Single Issue contains multiple unrelated problems or logical contradictions. (2) Undefined terminology: Uses unexplained jargon or abbreviations. (3) Unquantified requirements: Uses vague descriptions (e.g., "reasonable defaults", "user-friendly", "faster") without measurable acceptance criteria. (4) Low-quality test cases: Insufficient, overly broad, or missing test cases (passing the test does not prove the issue is resolved).
- *Excessive Reliance on External Resources:* (1) Core info relies on external links: Key descriptions/logs/steps are in links that may fail or be inaccessible. (2) Reliance on private repos: Reproduction depends on non-public codebases.

**Input Content**
Issue
{issue}

**Output Format**
Please provide your professional analysis and strictly output a number in the following format:
reason for evaluation: xxx
issue score: 0-10

*Figure 8.* The prompt template used for the deduction-based Issue quality evaluation.

*Table 7.* Statistics of the data collection and filtering process. **Total**: Raw data scraped from GitHub (2018–2025). **Filtered**: High-quality candidates remaining after applying strict quality and verifiability constraints.

| Language | Repositories | | Instances | |
|---|---|---|---|---|
| | **Total** | **Filtered** | **Total** | **Filtered** |
| Python | 9,515 | 1,722 | 91,072 | 60,872 |
| Java | 3,743 | 692 | 33,164 | 21,379 |
| Go | 3,563 | 771 | 61,216 | 41,113 |
| JavaScript | 7,388 | 1,128 | 18,952 | 13,864 |
| TypeScript | 4,839 | 1,295 | 58,637 | 41,753 |
| Rust | 1,743 | 467 | 20,106 | 13,500 |
| C | 2,273 | 443 | 2,965 | 2,057 |
| C++ | 3,061 | 672 | 11,283 | 7,227 |
| PHP | 1,962 | 427 | 14,129 | 8,376 |
| Ruby | 1,638 | 383 | 5,759 | 3,625 |
| **Total** | **39,725** | **8,000** | **317,283** | **213,766** |

Beyond domain categories, we also emphasize **Project Scale** to ensure the benchmark reflects the complexity of real-world software engineering. As illustrated in Figure 10, MEnvBench achieves a balanced distribution across both dimensions:

---

**Prompt for Repository Domain Classification**

**Task Definition**

Please classify the target repository into one of the 10 specific domain categories based on its metadata.

**Classification Categories**

- **1. Application Development:** Repositories focused on building end-user applications across web, mobile, and desktop platforms.
- **2. Database Systems:** Repositories implementing or interfacing with data storage and retrieval systems.
- **3. Data Science & Engineering:** Repositories for building data pipelines, processing large-scale data, and performing analytics.
- **4. Machine Learning & AI:** Repositories implementing machine learning algorithms, deep learning models, and AI applications.
- **5. Infrastructure Development:** Repositories for managing and automating cloud infrastructure and deployment workflows.
- **6. Specialized Programming Domain:** Repositories for domain-specific applications requiring specialized frameworks or engines.
- **7. Security Engineering:** Repositories focused on security, cryptography, and blockchain technologies.
- **8. UI/UX Engineering:** Repositories for building user interfaces and design systems.
- **9. Quality Assurance & Testing:** Repositories providing tools for testing, building, and ensuring code quality.
- **10. Embedded Systems & IoT:** Repositories for low-level programming, embedded systems, and Internet of Things applications.

**Input Format**

Please classify the following repository information:
```
Repository Name: {full_name}
Description: {description}
Primary Language: {language}
Topics: {topics}
```

**Output Format**

Please return **only** the classification result in JSON format:
```
{
   "category":  "Category Name",
   "reason":  "Brief explanation (1-2 sentences) based on description/topics"
}
```

**Classification Guidelines**

- **Primary Purpose:** Focus on the repository's main functionality.
- **Technology Stack:** Consider the primary language and tags.
- **Specificity:** If multiple categories apply, choose the most specific one.
- **Default Handling:** If information is insufficient, infer the most likely category from context.

*Figure 9.* The prompt template used for the LLM-based repository domain classification.

Figure 10(a) shows the coverage of 10 distinct application domains, while Figure 10(b) demonstrates that the dataset includes repositories of varying sizes, confirming its representativeness of substantial, complex codebases.

### D.5. Representative Data Instances from MEnvBench

To provide a comprehensive view of the MEnvBench data structure, we present representative task instances from both the Python and Java subsets. Listing 1 illustrates a Python sample sourced from the `home-assistant/core` repository, while Listing 2 depicts a Java task from `keycloak/keycloak`. These examples demonstrate the unified schema used across languages, capturing essential metadata, detailed problem statements, and the ground-truth verification patches.

*Listing 1.* A representative Python data instance from MEnvBench (Home Assistant). Long text fields (e.g., patches and problem statements) are truncated for brevity.

```
{
 "repo": "home-assistant/core",
 "pull_number": 104627,
 "instance_id": "home-assistant__core-104627",
 "issue_numbers": [
   45660
 ],
```

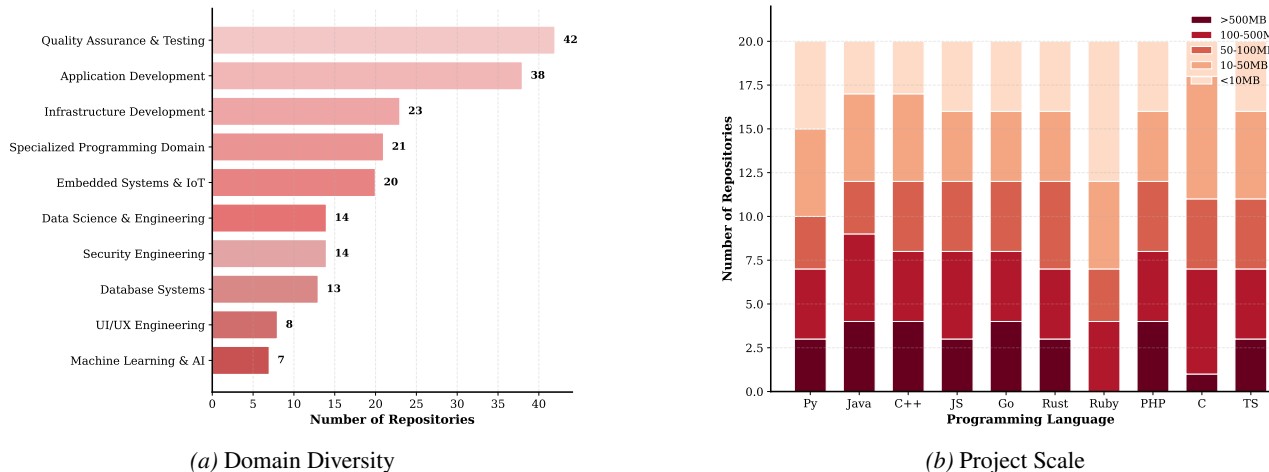

*(a)* Domain Diversity                    *(b)* Project Scale

*Figure 10.* **MEnvBench Diversity Statistics.** The dataset is analyzed across two key dimensions: (a) the distribution of repositories across 10 distinct application domains, and (b) the distribution of project scale.

```
"base_commit": "e594c19c1ecb9bc947b37d2af75e8d84f6a922e9",
"version": "0.38",
"language": "Python",
"created_at": "2023-11-28T00:01:38Z",
"commit_urls": [
  "https://github.com/home-assistant/core/commit/7bee0aea66673e92265106cee79efbb8582cd1f0"
],
"problem_statement": "Significant Change support for remote\nAdd [significant change](https://developers.home-
    assistant.io/docs/significant_change_index) support to the remote integration. All official properties need
    to be taken into consideration when deciding if a change is significant... [Content Truncated]",
"hints_text": "There hasn't been any activity on this issue recently. Due to the high number of incoming GitHub
    notifications... [Content Truncated]",
"all_hints_text": "There hasn't been any activity on this issue recently... [Content Truncated]",
"patch": "diff --git a/homeassistant/components/remote/significant_change.py b/homeassistant/components/remote/
    significant_change.py\nnew file mode 100644\nindex 0000000000000..8e5a36690411d\n--- /dev/null\n+++ b/
    homeassistant/components/remote/significant_change.py\n@@ -0,0 +1,27 @@\n+\"\"\"Helper to test significant
    Remote state changes.\"\"\"\n+from __future__ import annotations\n+\n+from typing import Any\n+... [Full
    content truncated for brevity]",
"test_patch": "diff --git a/tests/components/remote/test_significant_change.py b/tests/components/remote/
    test_significant_change.py\nnew file mode 100644\nindex 0000000000000..dcbfce213d65e\n--- /dev/null\n+++ b/
    tests/components/remote/test_significant_change.py\n@@ -0,0 +1,62 @@\n+\"\"\"Test the Remote significant
    change platform.\"\"\"\n+from homeassistant.components.remote import ATTR_ACTIVITY_LIST,
    ATTR_CURRENT_ACTIVITY\n+... [Full content truncated for brevity]"
}
```

*Listing 2.* A representative Java data instance from MEnvBench (Keycloak). This security-related task requires the agent to modify the secret generation logic to ensure sufficient entropy, involving changes to both utility classes and integration tests.

```
{
  "repo": "keycloak/keycloak",
  "pull_number": 39637,
  "instance_id": "keycloak__keycloak-39637_test",
  "issue_numbers": [
    38621
  ],
  "base_commit": "61fdfc2352a6e9da2e5dbeeb121cf731f48dfef9",
  "version": "2.5",
  "language": "Java",
  "created_at": "2025-05-12T10:47:31Z",
  "commit_urls": [
    "https://github.com/keycloak/keycloak/commit/aec69609309216a0535955714a93a3c7423e2f9e"
  ],
  "problem_statement": "Client secret generation provides lower than expected entropy\n### Describe the bug\nThe way
      how we generate client secrets in authentication flows... uses a character set consisting of 62 alphanumeric
      characters... For example, a 256-bit secret generated using 32 characters from a 62-character set results in
      only ˜192 bits of entropy... [Content Truncated]",
  "hints_text": "Changing to an enhancement and adding to sprint 67 for now...\nWe already use `SecureRandom` for
      generating random strings, but we can likely increase the character set and/or increase the length of the
      secret... [Content Truncated]",
  "all_hints_text": "Changing to an enhancement... [Content Truncated]",
  "patch": "diff --git a/common/src/main/java/org/keycloak/common/util/SecretGenerator.java b/common/src/main/java/
```

```
        org/keycloak/common/util/SecretGenerator.java\nindex ff73e855eeec..42eb86fb66d8 100644\n--- a/common/src/main
        /java/org/keycloak/common/util/SecretGenerator.java\n+++ b/common/src/main/java/org/keycloak/common/util/
        SecretGenerator.java\n@@ -70,4 +71,28 @@ public byte[] randomBytes(int length) {\n            return buf;\n
        }\n \n+    /**\n+     * Returns the equivalent length for a destination alphabet to have the same\n+     *
        entropy bits than a byte array random generated.\n+     */\n+    public static int equivalentEntropySize(int
        byteLengthEntropy, int dstAlphabetLeng) {\n+        return equivalentEntropySize(byteLengthEntropy, 256,
        dstAlphabetLeng);\n+    }\n... [Full content truncated for brevity]",
  "test_patch": "diff --git a/testsuite/integration-arquillian/tests/base/src/test/java/org/keycloak/testsuite/oauth
        /ClientAuthSecretSignedJWTTest.java b/testsuite/integration-arquillian/tests/base/src/test/java/org/keycloak/
        testsuite/oauth/ClientAuthSecretSignedJWTTest.java\nindex 6b91a1a01848..388c38447925 100644\n--- a/testsuite/
        integration-arquillian/tests/base/src/test/java/org/keycloak/testsuite/oauth/ClientAuthSecretSignedJWTTest.
        java\n+++ b/testsuite/integration-arquillian/tests/base/src/test/java/org/keycloak/testsuite/oauth/
        ClientAuthSecretSignedJWTTest.java\n@@ -290,16 +290,16 @@ private void processAuthenticateWithAlgorithm(
        String algorithm, Integer secretLe\n            configureDefaultProfileAndPolicy();\n \n        String
        firstSecret = clientResource.generateNewSecret().getValue(); //clientResource.getSecret().getValue();\n-
            assertThat(firstSecret.length(),is(secretLength));\n+        assertThat(firstSecret.length(), is(
        SecretGenerator.equivalentEntropySize(secretLength, SecretGenerator.ALPHANUM.length)));\n \n        //
        generate new secret, rotate the secret\n        String newSecret = clientResource.generateNewSecret().
        getValue();\n... [Full content truncated for brevity]"
}
```

## E. Detailed hyperparameter settings for MEnvAgent and baselines on MEnvBench.

All experiments were conducted on the MEnvBench benchmark. Unless otherwise specified, we used a fixed temperature of 0.5 and a global timeout of 3 hours per task to ensure fair comparison. The detailed hyperparameter configurations for MEnvAgent and all baseline methods are summarized in Table 8.

*Table 8.* Detailed hyperparameter settings for MEnvAgent and baselines on MEnvBench.

| Method | Parameter | Value |
|---|---|---|
| **Repo2Run** | Max Iterations | 50 |
| | Time Limit | 3 hours |
| | Temperature | 0.5 |
| | Concurrency | 15 |
| **SWE-Bench-Live** | Max Install Iterations | 20 |
| | Max verify Iterations | 20 |
| | Time Limit | 3 hours |
| | Temperature | 0.5 |
| | Concurrency | 15 |
| **SWE-Factory** | Max Iterations | 5 |
| | Time Limit | 3 hours |
| | Temperature | 0.5 |
| | Concurrency | 15 |
| **MEnvAgent (Ours)** | Max Iterations | 5 |
| | Time Limit | 3 hours |
| | Temperature | 0.5 |
| | Concurrency | 15 |

## F. Detailed Results on MEnvBench

**Performance.** Table 9 provides the comprehensive performance breakdown across all 10 programming languages evaluated in MEnvBench. The table reports Fail-to-Pass Rate (F2P), Pass Rate (PASS), and Time Cost (TIME) for all compared methods using both Kimi-K2 and Gemini-3-Flash backbones. The results indicate that MEnvAgent consistently achieves superior stability and resolution rates compared to baselines, particularly in complex system-level languages.

**Cost.** In addition to performance, we analyze the economic efficiency in Table 10. The data demonstrates that both MEnvAgent and SWE-Factory represent low-cost solutions, maintaining minimal token consumption per task compared to single-agent baselines that incur substantial overhead due to unplanned exploration. Although MEnvAgent incurs a marginally higher cost than SWE-Factory, this slight increment is well-justified by the significant gains in both F2P rate and time efficiency. Given that both methods operate within a highly affordable low-cost regime, this difference is negligible in practice and does not constitute a bottleneck for large-scale data expansion.

*Table 9.* Detailed performance comparison on MEnvBench across 10 languages. **F2P**, **Pass**, and **Time** denote Fail-to-Pass (%), Pass Rate (%), and Average Time Cost (s), respectively. "-" indicates the method is not applicable to the specific language. Our method is highlighted in **bold**.

| Method | Python | | | Go | | | Java | | | JavaScript | | | C | | |
|---|---|---|---|---|---|---|---|---|---|---|---|---|---|---|---|
| | F2P | PASS | TIME | F2P | PASS | TIME | F2P | PASS | TIME | F2P | PASS | TIME | F2P | PASS | TIME |
| *Kimi-K2* | | | | | | | | | | | | | | | |
| Repo2Run | 24 | 26 | 3112 | | - | | | - | | | - | | | - | |
| SWE-Bench-Live | 13 | 15 | 2589 | 3 | 6 | 4382 | 6 | 13 | 6231 | 11 | 16 | 4725 | | - | |
| SWE-Factory | 31 | 37 | 5512 | 50 | 57 | 5742 | 13 | 16 | 6182 | 37 | 40 | 5650 | 13 | 35 | 5777 |
| **MEnvAgent** | **53** | **60** | **2311** | **61** | **70** | **3171** | **28** | **33** | **4909** | **44** | **51** | **3210** | **13** | **39** | **3364** |
| *Gemini-3-Flash* | | | | | | | | | | | | | | | |
| Repo2Run | 27 | 32 | 3769 | | - | | | - | | | - | | | - | |
| SWE-Bench-Live | 21 | 23 | 2547 | 13 | 15 | 3689 | 8 | 10 | 5146 | 13 | 24 | 2441 | | - | |
| SWE-Factory | 44 | 51 | 5529 | 53 | 61 | 5193 | 18 | 24 | 6582 | 39 | 41 | 5690 | 18 | 37 | 5482 |
| **MEnvAgent** | **61** | **66** | **2530** | **61** | **66** | **3173** | **37** | **43** | **4797** | **48** | **63** | **3715** | **20** | **46** | **4598** |

| Method | C++ | | | Rust | | | TypeScript | | | PHP | | | Ruby | | |
|---|---|---|---|---|---|---|---|---|---|---|---|---|---|---|---|
| | F2P | PASS | TIME | F2P | PASS | TIME | F2P | PASS | TIME | F2P | PASS | TIME | F2P | PASS | TIME |
| *Kimi-K2* | | | | | | | | | | | | | | | |
| Repo2Run | | - | | | - | | | - | | | - | | | - | |
| SWE-Bench-Live | | - | | 11 | 15 | 7142 | 5 | 6 | 7867 | | - | | | - | |
| SWE-Factory | **7** | 21 | 5832 | 19 | 32 | 7773 | 23 | 27 | 8962 | 30 | 34 | 5759 | 39 | 46 | 6368 |
| **MEnvAgent** | **7** | **23** | **5254** | **33** | **42** | **4662** | **32** | **43** | **2330** | **37** | **40** | **2394** | **49** | **58** | **1782** |
| *Gemini-3-Flash* | | | | | | | | | | | | | | | |
| Repo2Run | | - | | | - | | | - | | | - | | | - | |
| SWE-Bench-Live | | - | | 21 | 25 | 5484 | 10 | 18 | 5187 | | - | | | - | |
| SWE-Factory | 6 | 24 | **5361** | 37 | 42 | 7311 | 24 | 28 | 8050 | 39 | 45 | 5969 | 55 | 62 | 6582 |
| **MEnvAgent** | **7** | **26** | 5995 | **43** | **57** | **4828** | **35** | **43** | **3632** | **40** | **47** | **2415** | **59** | **63** | **2399** |

*Table 10.* Cost analysis on MEnvBench across 10 languages. **Input** and **Output** represent token counts in thousands (**k**). **Cost** is the estimated average cost per task in **USD ($)**, calculated based on the pricing: Kimi-K2 ($0.6/1M In, $2.5/1M Out) and Gemini-3-Flash ($0.5/1M In, $1.5/1M Out). "-" indicates the method is not applicable to the specific language. Our method is highlighted in **bold**.

| Method | Python | | | Go | | | Java | | | JavaScript | | | C | | |
|---|---|---|---|---|---|---|---|---|---|---|---|---|---|---|---|
| | Input | Output | Cost | Input | Output | Cost | Input | Output | Cost | Input | Output | Cost | Input | Output | Cost |
| *Kimi-K2* | | | | | | | | | | | | | | | |
| Repo2Run | 630 | 2 | 0.38 | | - | | | - | | | - | | | - | |
| SWE-Bench-Live | 1220 | 36 | 0.82 | 640 | 47 | 0.50 | 770 | 37 | 0.55 | 710 | 21 | 0.48 | | - | |
| SWE-Factory | 90 | 12 | 0.08 | 65 | 7 | 0.06 | 102 | 10 | 0.09 | 72 | 8 | 0.06 | 114 | 14 | 0.10 |
| MEnvAgent | **141** | **9** | **0.11** | **116** | **7** | **0.09** | **130** | **10** | **0.10** | **167** | **8** | **0.12** | **198** | **13** | **0.15** |
| *Gemini-3-Flash* | | | | | | | | | | | | | | | |
| Repo2Run | 890 | 12 | 0.46 | | - | | | - | | | - | | | - | |
| SWE-Bench-Live | 2010 | 228 | 1.35 | 1300 | 144 | 0.87 | | - | | 1900 | 149 | 1.17 | | - | |
| SWE-Factory | 86 | 42 | 0.11 | 77 | 42 | 0.10 | 110 | 54 | 0.14 | 65 | 41 | 0.09 | 283 | 85 | 0.27 |
| MEnvAgent | **211** | **104** | **0.26** | **148** | **111** | **0.24** | **192** | **205** | **0.40** | **166** | **224** | **0.42** | **315** | **212** | **0.48** |

| Method | C++ | | | Rust | | | TypeScript | | | PHP | | | Ruby | | |
|---|---|---|---|---|---|---|---|---|---|---|---|---|---|---|---|
| | Input | Output | Cost | Input | Output | Cost | Input | Output | Cost | Input | Output | Cost | Input | Output | Cost |
| *Kimi-K2* | | | | | | | | | | | | | | | |
| Repo2Run | | - | | | - | | | - | | | - | | | - | |
| SWE-Bench-Live | | - | | 580 | 12 | 0.38 | 580 | 45 | 0.46 | | - | | | - | |
| SWE-Factory | 107 | 13 | 0.10 | 111 | 11 | 0.09 | 61 | 8 | 0.06 | 79 | 9 | 0.07 | 62 | 7 | 0.05 |
| MEnvAgent | **187** | **15** | **0.15** | **97** | **6** | **0.07** | **106** | **7** | **0.08** | **122** | **8** | **0.09** | **92** | **7** | **0.07** |
| *Gemini-3-Flash* | | | | | | | | | | | | | | | |
| Repo2Run | | - | | | - | | | - | | | - | | | - | |
| SWE-Bench-Live | | - | | 1170 | 134 | 0.79 | | - | | | - | | | - | |
| SWE-Factory | 224 | 76 | 0.23 | 77 | 44 | 0.10 | 71 | 45 | 0.10 | 73 | 46 | 0.11 | 110 | 54 | 0.14 |
| MEnvAgent | **282** | **286** | **0.57** | **136** | **77** | **0.18** | **151** | **134** | **0.28** | **182** | **180** | **0.36** | **157** | **168** | **0.33** |

## G. Cross-Language Ablation Study on Environment Reuse

To thoroughly evaluate the impact of the environment reuse mechanism across a polyglot landscape, we conducted an extensive cross-language ablation study. We curated a representative subset of MEnvBench, spanning 10 distinct programming languages, 4 repositories per language, and 5 instances per repository, culminating in a total of 200 tasks.

*Table 11.* Cross-language ablation study on environment reuse across 10 programming languages (200 tasks in total). RSR, PASS, and TIME denote the Repository-level Success Rate, Instance-level Pass Rate, and Total Execution Time (seconds), respectively.

| Language | RSR (%) | PASS (%) | | Time (s) | |
|---|---|---|---|---|---|
| | | MEnvAgent | *w/o* Reuse | MEnvAgent | *w/o* Reuse |
| Python | 30.0 | 70.0 | 60.0 | 2861 | 3994 |
| Ruby | 30.0 | 65.0 | 50.0 | 2364 | 3776 |
| TypeScript | 25.0 | 60.0 | 55.0 | 3826 | 4850 |
| Go | 25.0 | 55.0 | 45.0 | 4309 | 5117 |
| Rust | 25.0 | 55.0 | 50.0 | 4698 | 5639 |
| Java | 20.0 | 45.0 | 40.0 | 4322 | 5141 |
| JavaScript | 20.0 | 45.0 | 30.0 | 3916 | 5275 |
| PHP | 15.0 | 30.0 | 25.0 | 3212 | 3764 |
| C++ | 15.0 | 20.0 | 10.0 | 5377 | 6238 |
| C | 10.0 | 20.0 | 15.0 | 5804 | 6455 |
| **Average** | **21.5** | **46.5** | **38.0** | **4069** | **5025** |

**Results.** As compiled in Table 11, incorporating the environment reuse mechanism yields an average Reuse Success Rate (RSR) of 21.5% and an +8.5% absolute improvement in the instance-level PASS rate across all 10 languages. Crucially, this performance gain is not merely a reflection of the global average, but holds consistently on a per-language basis across the entire polyglot spectrum, fully demonstrating the universal and consistent efficacy of our reuse mechanism. Furthermore, the reuse mechanism considerably optimizes computational efficiency, slashing the average overall execution time by approximately 19%.

## H. Detailed Failure Analysis on C++ Tasks

As indicated in Table 9, compiled languages, especially C++ and C, present the most grueling bottlenecks for automated environment construction across different base foundation models. For example, even when powered by the advanced Gemini-3-Flash, MEnvAgent only achieves a 26% PASS rate in C++ and a 23% PASS rate when using Kimi-K2, lagging significantly behind dynamic languages like Python (66%). To diagnose these persistent pain points, we conducted a meticulous failure analysis based on the comprehensive compilation and execution failures encountered within our MEnvBench tasks. The primary failure modes are categorized and quantified as follows.

- **Out-of-Memory (OOM) During Compilation (22%):** Heavy C++ template expansions and parallel compilation jobs (e.g., `make -j` or `ninja`) frequently exhaust the hard memory ceilings assigned to the containerized sandboxes before any executable test binary can be produced. A representative case is observed in `godotengine/godot`, where compiler processes are abruptly killed by the OS kernel due to memory starvation.

- **Non-Standard Build System Incompatibility (20%):** Unlike ecosystems with unified package/build managers (e.g., Cargo for Rust, Go modules), advanced C++ projects extensively utilize diverse or bespoke build orchestrators like Bazel, Nix, SCons, or Colcon (e.g., `RobotLocomotion/drake`). These systems require highly specific, contextual environment setup logic that LLM agents struggle to infer under a rigid token and iteration budget.

- **Heavy Dependency Installation Failure (18%):** Large-scale robotic or graphics frameworks such as ROS2 (`moveit/moveit2`) pull in gigabytes of upstream prerequisites. This frequently triggers `apt` package manager hangs, transient network drops, upstream repository version conflicts, or broken PPA mirrors, which derails the automated pipeline.

- **Missing System Development Libraries (8%):** Many C++ test suites depend implicitly on low-level system shared libraries (e.g., `liblld-dev`, `Qt6Keychain`) that are missing from vanilla base Docker images. Because package

naming conventions fluctuate drastically across Linux distributions and releases, the agent often fails to map the missing linker error to the correct package name.

- **Network Download Timeout (5%):** Fetching massive external third-party binaries or Git submodules during the configure stage occasionally hangs indefinitely, blocking subsequent compilation and test verification steps.

These diagnostic insights highlight that the primary bottlenecks for automated polyglot environment building remain heavily structural and infrastructural rather than purely algorithmic. In future work, we aim to design more flexible resource-allocating sandboxes and specialized toolsets to mitigate these systemic failures, and we plan to share these robust environments with the open-source community.

# I. Details of Scaling Verifiable SWE Datasets

In this section, we describe how we leveraged the **MEnvAgent** framework to scale up the production of verifiable environments and training trajectories, resulting in the **MEnvData-SWE** dataset.

## I.1. Dataset Construction Pipeline

Our pipeline builds upon the high-quality candidate pool established in Appendix D. Utilizing the filtered repositories and instances presented in Table 7, we focus on the subsequent stages: Environment Construction via MEnvAgent and Execution-based Verification.

## I.2. Environment Construction

This phase constitutes the core of our pipeline, where **MEnvAgent** is deployed to automatically retrieve codebases, resolve dependencies, and generate Docker images.

Environment construction is the most computationally intensive and time-consuming component of the pipeline, serving as the primary performance bottleneck. On standard local infrastructure, the Docker daemon severely limits concurrent build operations; our preliminary benchmarks indicate that effective concurrency is capped at approximately 10–15 tasks. Given that resolving complex dependencies for legacy repositories can span several hours per instance, achieving large-scale dataset generation on local machines is virtually infeasible.

To overcome this scalability barrier, we re-engineered the underlying infrastructure to support **Kubernetes (K8s)** orchestration. By decoupling the build agents from the host limits, we achieved (1,000+ parallel builds). This architectural shift enabled the rapid construction of thousands of environment-aware images in a short timeframe. We commit to open-sourcing this high-throughput build infrastructure and the resulting data to accelerate community research in large-scale software engineering.

## I.3. Execution-based Verification

Once the environments are constructed, we verify their validity to ensure they represent genuine, reproducible bug-fix scenarios.

**Fail-to-Pass (F2P).** We implement a rigorous *Fail-to-Pass* verification protocol using test cases extracted from the original Pull Requests. A task is deemed a valid SWE task only if it satisfies the following two-stage strict check:

1. **Reproduction Phase (Fail):** The test script is executed in the environment with only the *Test Patch* applied (simulating the buggy state). The outcome must be a **Failure**, confirming that the reported issue is reproducible within the environment.

2. **Verification Phase (Pass):** The test script is executed in the environment with both the *Test Patch* and the *Fix Patch* applied (simulating the fixed state). The outcome must be a **Success**, confirming that the provided patch effectively resolves the issue.

Only instances that survive this rigorous pipeline are included in the final dataset, guaranteeing that every sample is grounded in a reproducible, executable environment.

## I.4. Details of MEnvData-SWE Dataset

In this section, we provide a detailed statistical breakdown of the **MEnvData-SWE** dataset, which was constructed using the MEnvAgent pipeline and utilized for the Supervised Fine-Tuning (SFT) experiments described in Section 6.3.

Table 12 presents the distribution of data across different programming languages. The dataset metrics are defined as follows:

- **Repos:** The number of unique source repositories sourced from GitHub.

- **Instances:** The number of unique task instances (comprising a specific issue and a corresponding pull request snapshot).

- **Trajectories:** The number of successfully verified agent interaction traces (Thought-Action sequences) collected via rejection sampling. These trajectories serve as the high-quality instruction data for model fine-tuning.

As shown in the table, the dataset exhibits a diverse distribution across ecosystems. While popular languages like Rust and JavaScript contribute a significant portion of the data due to their active open-source communities, the dataset also maintains coverage for lower-level languages like C and enterprise-heavy languages like Java and PHP, ensuring multilingual generalization for trained models.

*Table 12*. **Detailed Breakdown.** Language-specific statistics of the MEnvData-SWE dataset. The table details the count of unique repositories, task instances, and verifiable solution trajectories collected for each language.

| Language | # Repos | # Instances | # Trajectories |
|---|---|---|---|
| Go | 169 | 502 | 502 |
| Java | 14 | 33 | 47 |
| JavaScript | 119 | 578 | 694 |
| PHP | 21 | 159 | 395 |
| Python | 192 | 477 | 543 |
| Ruby | 79 | 283 | 749 |
| Rust | 249 | 769 | 769 |
| TypeScript | 76 | 157 | 157 |
| C | 9 | 16 | 16 |
| C++ | 14 | 31 | 0 |
| **Total** | **942** | **3,005** | **3,872** |

## I.5. Comparison with Other Verifiable Datasets

To contextualize the scale and diversity of our contribution, we compare **MEnvData-SWE** against prominent verifiable SWE benchmarks and training datasets in Table 13.

Existing benchmarks, while pivotal, are often restricted to single-language ecosystems (primarily Python) or lack repository diversity. similarly, recent training datasets typically rely on synthetic mutations or generated tests (e.g., SWE-Smith, SWE-Flow), lacking the fidelity of real-world development scenarios. In contrast, MEnvData-SWE distinguishes itself by simultaneously achieving high polyglot coverage and grounding all data in authentic, user-submitted issues.

**Conclusion.** As demonstrated by the comparison, MEnvData-SWE represents **the largest open-sourced polyglot dataset of realistic verifiable Docker environments to date**, alongside solution trajectories that enable consistent performance gains on SWE tasks across a wide range of models.

## I.6. Training Implementation Details

We fine-tune all models using a unified supervised fine-tuning (SFT) framework based on Megatron-LM. To equip the models with long-context capabilities, we scale the training sequence length to 128k tokens ($131,072$).

*Table 13.* **Comparative Overview.** Comparison of MEnvData-SWE with existing verifiable SWE benchmarks and training datasets. **Langs**: Number of supported languages. **# Repos**: Number of source repositories. **# Instances**: Number of task instances. **# Trajectories**: Number of agent solution trajectories. **Realistic**: Originates from real-world issues (✓) vs. synthetic/generated (✗).

| Dataset | Langs | # Repos | # Instances | # Trajectories | Realistic |
|---|---|---|---|---|---|
| *Verifiable SWE Benchmarks* | | | | | |
| SWE-bench Verified | 1 | 12 | 500 | - | ✓ |
| SWE-bench Multilingual | 9 | 42 | 300 | - | ✓ |
| Multi-SWE-bench | 7 | 39 | 1,632 | - | ✓ |
| SWE-bench Multimodal | 1 | 17 | 617 | - | ✓ |
| FEA-Bench | 1 | 83 | 1,401 | - | ✓ |
| SWE-Perf | 1 | 12 | 140 | - | ✓ |
| *Verifiable SWE Training Datasets* | | | | | |
| SWE-Factory | 1 | - | - | 2,809 | ✓ |
| SWE-Smith | 1 | 128 | 50,000 | 5,016 | ✗ |
| SWE-Flow | 1 | 62 | 16,061 | - | ✗ |
| SWE-Synth | 1 | 7 | 9,459 | 3,018 | ✗ |
| SWE-Mirror | 1 | 40 | 60,000 | 6,431 | ✗ |
| **MEnvData-SWE (Ours)** | **10** | **942** | **3,005** | **3,872** | ✓ |

**Optimization and Hyperparameters.** We fine-tune the models on a dataset of approximately 4k instances for 3 epochs. We employ the AdamW optimizer with a global batch size of $8$. The learning rate is scheduled with a constant strategy, where the peak learning rate is set to $1 \times 10^{-5}$ and decays to a minimum of $1 \times 10^{-9}$, with no warmup steps. We set the gradient clipping norm to $1.0$ and use a weight decay of $0.1$. For MoE-based models, we apply an auxiliary loss with a coefficient of $1 \times 10^{-3}$ to ensure load balancing among experts.

**Infrastructure and Efficiency.** To efficiently train large-scale models with such long contexts, we utilize a comprehensive 3D parallelism strategy deployed on NVIDIA H800 GPUs, configuring Tensor Parallelism (TP), Pipeline Parallelism (PP), and Expert Parallelism (EP) all to size $8$, alongside Sequence Parallelism. Furthermore, we leverage Flash Attention 2 to accelerate attention computation and enable full activation checkpointing (recompute) to significantly reduce memory fragmentation during training.

