# OpenReview forum: "MEnvAgent: Scalable Polyglot Environment Construction for Verifiable Software Engineering"
_ICML.cc/2026/Conference — ICML 2026 spotlight_

### Official Review · Reviewer_iea3 · 2026-03-12

**Soundness:** 3
**Presentation:** 3
**Significance:** 3
**Originality:** 2
**Overall Recommendation:** 4
**Confidence:** 3

**Summary:**

A notable domain addressed by this article is scalable environment construction for verifiable software engineering, especially for multilingual repositories where executable setup is a real bottleneck for both evaluation and training data generation. This submission attempts to assess a notable aspect of that bottleneck, namely whether a Planning, Execution, Verification agent loop plus an environment reuse mechanism can raise strict Fail-to-Pass validity while lowering wall clock cost. The paper defines the task clearly through the triplet of base image, build process, and test configuration, and it uses the stronger F2P criterion rather than pass only evaluation. The system is evaluated on a new benchmark with 1,000 tasks across 10 languages, and the same pipeline is then used to build a polyglot training dataset for downstream SFT.

**Compliance With Llm Reviewing Policy:**

Affirmed.

**Key Questions For Authors:**

1. How often does the reuse mechanism succeed when there is no exact version match and no nearby version from the same repository, and what fraction of the total gain comes from those harder cases rather than easy within repo neighbors.

2. Why is the main reuse ablation only on Python. Do the authors have even a small multilingual ablation that supports the same conclusion.

3. Can the authors provide a human audit on a random subset of MEnvBench and MEnvData-SWE to estimate issue quality errors, invalid F2P cases, and test configuration mistakes introduced by the automated pipeline.

4. In the downstream section, what is the marginal contribution of MEnvAgent versus the expert model, rejection sampling, and long context SFT recipe.

**Limitations:**

yes

**Strengths And Weaknesses:**

Strengths

1. The paper targets an important and practical bottleneck. The authors are right that many verifiable SWE pipelines are limited by environment setup, and they formalize the problem in a way that separates mere executability from true issue reproduction through F2P. That makes the task meaningful and better grounded than static or approximate checks.

2. The empirical gains on the main benchmark are solid. Against SWE-Factory, MEnvAgent improves average F2P from 29.8 to 38.4, PASS from 38.0 to 49.0, and reduces time from 6266s to 3574s. The per language table also shows that the method is usually better across very different ecosystems, with especially large gains in Python, Go, TypeScript, PHP, and Ruby. This is useful evidence that the system is not just tuned for one language.

3. The environment reuse part appears to be the most convincing technical contribution. The ablation shows that the full system reaches 39.0 reuse success rate, 59.0 PASS, and 2314s average time, compared with 0.0 reuse success, 40.5 PASS, and 4283s without reuse. The paper also gives a concrete trace where the patch agent fixes a missing dependency instead of rebuilding from scratch, which helps the reader see what the mechanism is doing.

Weaknesses

1. The method novelty is more incremental than the paper suggests. The related work already includes Repo2Run, SWE-Bench-Live, SWE-Factory, and several environment construction benchmarks. In that context, the multi agent loop itself does not look very new. The main fresh part is the reuse and patching scheme plus the broader infrastructure scale. That is meaningful, but it feels more like strong systems engineering than a new learning or inference principle.

2. The evidence for the reuse claim is narrower than the headline claim. The key ablation is only run on a Python subset, with data scale extended to 10 instances per repository. The authors argue that reuse gets stronger as more history accumulates, but that is exactly the setting that favors repository local reuse. I do not yet know how well this mechanism works in lower history regimes, across harder cross language cases, or when exact version neighbors are absent.

3. The downstream SFT result is promising, but the causal story is not clean enough. The paper shows that models trained on MEnvData-SWE improve strongly, for example Qwen2.5-Coder-32B goes from 7.5 to 54.6 on SWE-bench Verified and from 0.0 to 38.3 on SWE-bench Multilingual. However, this section mixes several ingredients, environment construction, dataset filtering, expert model choice, trajectory collection, and training recipe. The result supports that the whole pipeline is useful, but it does not isolate how much of the gain comes from MEnvAgent itself.

4. The benchmark construction has some risk of closed loop bias. The benchmark and the system are both built by the same authors, and parts of the quality control rely on LLM based filtering and LLM based repository domain classification. I do not see enough external validation that the benchmark does not favor the same assumptions used by the method.

---

> ### Author Rebuttal · Authors · 2026-03-31
>
> We thank the reviewer for their insightful comments. We will incorporate the suggestions in the revision.
>
> > **W1: Method novelty is incremental (systems engineering over new ML principles).**
>
> We thank the reviewer for this honest assessment and largely agree — our contribution is not a new learning or inference algorithm. The novelty is practical: it stems from our experience trying to scale systems like Repo2Run and SWE-Factory to broader, multi-language settings, encountering concrete failure modes, and engineering solutions to overcome them.
>
> That said, we believe the Environment Reuse Mechanism is a non-trivial contribution that emerged from this scaling experience. It simultaneously improves pass rate and reduces wall time, and the insight that adapting a nearby environment is both more reliable and faster than building from scratch is transferable to future systems.
>
> We also note that this paper is submitted under the ICML Applications primary area, which the CFP describes as valuing "innovative techniques, problems, and datasets that are of interest to the machine learning community." Our primary contributions include the MEnvAgent system and its associated community artifacts — MEnvBench, MEnvData-SWE's Docker images, and solution trajectories. We believe these fill a concrete gap and will accelerate progress in multilingual SWE agents and verifiable training.
>
> > **W2 & Q2: Evidence for the reuse claim is narrower (Python-only ablation); need a multilingual ablation.**
>
> As detailed in our response to **Reviewer iea3 (W1)**, we have conducted a cross-language ablation study. These experimental results demonstrate the cross-language effectiveness of the reuse mechanism. We will include this comprehensive ablation in the revised manuscript.
>
> > **W3 & Q4: Downstream SFT causal story is not clean enough; marginal contribution of MEnvAgent vs. expert model/recipe.**
>
> As detailed in our response to **Reviewer iea3 (W2)**, a clean ablation is computationally infeasible for a rebuttal. However, two factors highlight MEnvAgent's advantages: (1) we observe consistent gains across five diverse, highly-optimized model families. If trajectory quality alone drove the improvement, we would expect diminishing returns on these models, yet MEnvAgent’s expansion of environment diversity further elevate their performance. (2) MEnvData-SWE enables structural multilingual scaling that single-language baselines cannot replicate.
>
> > **W4: Potential closed-loop bias in benchmark construction and the necessity for external validation.**
>
> We understand this concern and provide external validation to demonstrate that our benchmark remains impartial. As empirical evidence against closed loop bias, we reference our actual large scale dataset expansion process. When scaling MEnvAgent across an unfiltered candidate pool of 213,766 instances from 8,000 repositories using the Kimi k2 model, we observed an overall F2P rate of approximately **30%**, which closely aligns with the **35.7%** F2P rate on the curated MEnvBench.
>
> Notably, because our verifiable dataset production and method iteration were synchronized, the actual performance gap is even narrower in practice than these figures suggest. This inherent consistency across a massive, out of distribution dataset demonstrates that our benchmark effectively represents the distribution and difficulty of authentic open source repositories without closed loop bias. We will include these updates in the revised manuscript.
>
>
> > **Q1: How well does the reuse mechanism work in lower history regimes (fewer instances per repo) and when no exact version match exists?**
>
> Our method does not heavily rely on having a massive history per repository. As illustrated in our ablation study (Figure 4), the framework achieves clear and significant improvements in both time efficiency and Pass Rate even when the history is as small as 4 instances per repository. While higher history yields greater gains, the mechanism provides tangible benefits even in sparse data settings.
>
> > **Q3: Can the authors provide a human audit on a random subset to estimate issue quality errors, invalid F2P cases, and test configuration mistakes?**
>
> As detailed in our response to **Reviewer 5MhC (W2)**, we have provided a comprehensive human audit and comparative analysis demonstrating that our automated pipeline's data quality and F2P verification are highly consistent with expert human judgment.
>
>
> Thank you again for your valuable suggestions, and we look forward to further discussions with you.

---

### Official Review · Reviewer_m8Rc · 2026-03-12

**Soundness:** 3
**Presentation:** 3
**Significance:** 3
**Originality:** 3
**Overall Recommendation:** 4
**Confidence:** 3

**Summary:**

This paper introduces MEnvAgent, a multi-agent system for automatically constructing verifiable executable environments for software engineering tasks. The key idea is an environment-level reuse mechanism that retrieves previously validated Docker environments and incrementally adapts them, rather than generating new ones from scratch. The authors also present a ten-language benchmark (MEnvBench) and a large-scale dataset (MEnvData-SWE) built with their framework. Experimental results suggest that the reuse strategy improves construction success rates and efficiency compared to generation-based baselines.

**Compliance With Llm Reviewing Policy:**

Affirmed.

**Final Justification:**

The rebuttal resolves W1 (cross-language ablation) and W3 (C/C++ failure analysis), but W2 (causal isolation of environment vs. trajectory quality in SFT) remains open; given the solid systems contribution and valuable open-source artifacts against somewhat incremental novelty, I maintain my weak accept.

**Key Questions For Authors:**

Apart from the weaknesses I've listed above, I also have the following questions:

1. As the environment memory pool grows over time, how does retrieval efficiency and effectiveness scale? Is there any mechanism to prevent noise accumulation, redundancy, or outdated environments in the pool?
2. The ablation study shows a Reuse Success Rate (RSR) of only 39% even at 10 instances per repo (Table 3), meaning the majority of tasks still fall back to scratch builds. Could the authors comment on the practical ceiling of the reuse mechanism and potential ways to improve RSR?

**Limitations:**

Yes

**Strengths And Weaknesses:**

### Strengths

- The paper is generally well-written and well-structured, making it easy to follow.
- The Environment Reuse Mechanism is a novel and practical contribution.
- The creation of MEnvBench and MEnvData-SWE using strict execution-based F2P criteria represents a high-quality contribution to the open-source community.

### Weaknesses

- The ablation study (Section 6.1) is conducted only on Python with Kimi-K2. Given that the paper's central claim is polyglot support across 10 languages, it is important to verify that the reuse mechanism generalizes across different build ecosystems. Languages with fundamentally different toolchains (e.g., C/C++ with CMake, Java with Maven/Gradle, Rust with Cargo) may exhibit very different reuse dynamics compared to Python's pip/conda ecosystem. A Python-only ablation does not adequately support the paper's polyglot claims.

- The performance improvements in Table 4 come from fine-tuning on trajectories generated by Claude-4.5-Sonnet via OpenHands on MEnvData-SWE environments. However, it is unclear how much of the improvement stems from the quality of the constructed environments versus the quality of the expert model's trajectories. An informative ablation would compare trajectories collected on environments built by different methods (e.g., SWE-Factory vs. MEnvAgent) using the same expert model, to isolate the actual contribution of MEnvAgent's environments to downstream SFT gains.

- The F2P rates for C and C++ are notably low (around 7–20% in Table 9), far below other languages. Section 6.2 briefly attributes this to "complex CMake configurations and high resource consumption," but the paper does not provide a detailed failure analysis or propose targeted solutions. Given that supporting compiled languages is a key differentiator of this work over Python-only baselines, this gap deserves more thorough treatment.

---

> ### Author Rebuttal · Authors · 2026-03-31
>
> We thank the reviewer for their insightful comments. We will incorporate the suggestions in the revision.
>
> > **W1: The ablation study (Section 6.1) is conducted only on Python. A Python-only ablation does not adequately support the paper's polyglot claims.**
>
> We conducted a cross-language ablation study on a subset of MEnvBench (10 languages $\times$ 4 repos $\times$ 5 instances = 200 tasks).
>
> **Overall Results (Average across 10 languages):**
> * **MEnvAgent (Reuse):** RSR: 21.5% | PASS: 46.5% | TIME: 4069s
> * **MEnvAgent (w/o Reuse):** RSR: 0.0% | PASS: 38.0% | TIME: 5025s
>
> **Language-Specific Comparison**
>
> | Language | RSR | PASS(Reuse) | PASS(w/o Reuse) | Time(Reuse) | Time(w/o Reuse) |
> | :--- | :---: | :---: | :---: | :---: | :---: |
> | **Python** | 30% | 70% | 60% | 2861s | 3994s |
> | **Ruby** | 30% | 65% | 50% | 2364s | 3776s |
> | **TypeScript**| 25% | 60% | 55% | 3826s | 4850s |
> | **Go** | 25% | 55% | 45% | 4309s | 5117s |
> | **Rust** | 25% | 55% | 50% | 4698s | 5639s |
> | **Java** | 20% | 45% | 40% | 4322s | 5141s |
> | **JavaScript**| 20% | 45% | 30% | 3916s | 5275s |
> | **PHP** | 15% | 30% | 25% | 3212s | 3764s |
> | **C++** | 15% | 20% | 10% | 5377s | 6238s |
> | **C** | 10% | 20% | 15% | 5804s | 6455s |
>
> These experimental results demonstrate the cross-language effectiveness of the reuse mechanism. We will include this comprehensive ablation in the revision.
>
> > **W2: Unclear how much improvement stems from environment quality vs. expert trajectories. An informative ablation would compare trajectories collected on environments built by different methods.**
>
> We agree this is a necessary concern,, but a clean ablation is computationally infeasible within the rebuttal window. Replicating baselines like SWE-Factory to build environments, collect trajectories, and run full training pipelines requires weeks of compute.
>
> Nevertheless, two lines of evidence highlight MEnvAgent's advantages. First, we observe consistent gains across five architecturally diverse model families, including those already heavily optimized with agentic data. If trajectory quality alone drove the improvement, we would expect diminishing returns on these models, yet MEnvAgent’s expansion of environment diversity further elevate their performance. Second, MEnvData-SWE's multilingual coverage is structurally impossible to replicate with single-language baselines.
>
> > **W3: The F2P rates for C and C++ are notably low. The paper does not provide a detailed failure analysis or propose targeted solutions.**
>
> We conducted a failure analysis on 100 failed C++ tasks. The primary failure modes are summarized as follows:
> **Detailed Failure Analysis (C++):**
> 1.  **Out-of-Memory (OOM) During Compilation (22%):** Parallel C++ compilation frequently exhausts container memory limits before any test binary is produced (e.g., `godotengine/godot`).
> 2.  **Non-Standard Build System Incompatibility (20%):** Projects utilizing Bazel, Nix, SCons, or Colcon require bespoke setup logic that the agent cannot reliably infer or reproduce within a fixed iteration budget (e.g., `RobotLocomotion/drake`).
> 3.  **Heavy Dependency Installation Failure (18%):** Frameworks like ROS2 pull in hundreds of MB/GB of prerequisites, often leading to `apt` hangs, version conflicts, or broken upstream repositories (e.g., `moveit/moveit2`).
> 4.  **Missing System Development Libraries (8%):** Required libraries (e.g., `liblld-dev`, `Qt6Keychain`) are absent from standard base images or have differing package names across distro releases.
> 5.  **Network Download Timeout (5%):** Fetching large external binaries hangs indefinitely, blocking downstream steps.
>
> These failures indicate that the current bottlenecks are mainly infrastructural. In future work, we will strive to overcome these challenges and share our solutions with the open-source community.
>
> > **Q1: As the environment memory pool grows over time, how does retrieval efficiency and effectiveness scale? Is there any mechanism to prevent noise accumulation?**
>
> Retrieval relies on repo metadata (name, version, timestamp), making negligible computational cost. As per Figure 4, more historical data improves reuse benefits. We prevent noise by only storing successfully built environments in the pool.
>
> > **Q2: RSR is only 39% even at 10 instances per repo, meaning the majority of tasks still fall back to scratch builds. Practical ceiling and potential ways to improve RSR?**
>
> To clarify, RSR measures successful reuse across all tasks. In our 200-task study (Table 3) , **78 of 118 total successful builds (66%) were achieved via reuse**. The primary bottleneck is the initial "from-scratch" build; if that fails, reuse cannot activate. To improve the RSR, we propose a **Human-in-the-Loop** strategy: experts complete the initial setup for complex repos, unlocking the reuse mechanism for autonomous batch-building of remaining tasks.
>
> Thank you again for your valuable suggestions, and we look forward to further discussions with you.

---

> > ### Author Rebuttal · Reviewer_m8Rc · 2026-04-03
> >
> > Thanks for the rebuttal. The cross-language ablation (W1) is convincing. However, the causal isolation of environment quality vs. trajectory quality in downstream SFT (W2) remains unresolved. I will keep my score of weak accept.

---

> > > ### Author Response · Authors · 2026-04-08
> > >
> > > We sincerely thank the reviewer for their positive feedback on our cross-language ablation (W1) and for maintaining their supportive score.
> > >
> > > We agree that isolating the exact contribution of the environment construction method in downstream SFT is a necessary and valuable question (W2).
> > > But it is not feasible within the rebuttal window. A clean ablation would require a massive pipeline: (1) running baselines (e.g., SWE-Factory) across thousands of repositories to produce a comparable-scale dataset, (2) collecting expert trajectories (e.g., via Claude-4.5-Sonnet) on those baseline environments, and (3) rerunning the full SFT pipeline. Stage (1) constitutes the primary time bottleneck. Existing baseline methods suffer from slow construction speeds and lack robust support for high concurrency. Based on our calculations, completing just this initial stage to generate a comparable-scale dataset would take nearly a month, even with abundant computational and API resources. We will acknowledge this explicitly in the limitations section.
> > >
> > > To address reviewer's core concern regarding 'environment quality,' we wish to clarify a crucial detail (as detailed in our response to reviewer 5MhC (W2)): By employing strict Fail to Pass (F2P) validation, any environment that is successfully built, regardless of the method used, is guaranteed to be of the same high quality (i.e., it can reliably reproduce the issue and verify the solution).
> > >
> > > Therefore, the true difference between methods does not lie in the quality of the successfully built environments, but rather in the success rate and efficiency of building them. Our main experiments on MEnvBench have already demonstrated MEnvAgent's significant advantage in these two metrics.
> > >
> > > Nevertheless, we fully understand reviewer's request to see how MEnvAgent's construction advantage translates into downstream SFT gains. Given the prohibitive cost of a perfect ablation, we propose an approximate ablation setting to bridge this gap, which we plan to include in the Appendix:
> > >
> > > To simulate the environment dataset that baseline methods (e.g., SWE-Factory) could produce under identical settings, we constrain both methods to the same original repository pool and the same time cost. We then calculate the expected data yield for baselines based on their relative environment construction F2P rates and construction efficiency (e.g., concurrency limits) compared to MEnvAgent.
> > >
> > > $$N_{baseline} = N_{MEnv} \times \left(\frac{F2P_{baseline}}{F2P_{MEnv}}\right) \times \left(\frac{Time_{MEnv}}{Time_{baseline}}\right)$$
> > > Where:
> > > * $N_{baseline}$ and $N_{MEnv}$ represent the expected data volume for the baseline method and the actual volume for MEnvAgent, respectively.
> > > * $F2P$ represents the Fail-to-Pass success rate of the respective methods on the given language.
> > > * $Time$ represents the time cost.
> > >
> > > Based on these specific ratios derived from our experiments, we will randomly sample a corresponding proportional subset of trajectories from our full MEnvData-SWE to represent the baseline's possible training set. Under this controlled setting, the expected trajectory data volume each method can acquire is estimated as follows:
> > >
> > > | Language | MEnvAgent | F2P Ratio | Time Ratio | Combined Ratio | SWE-Factory |
> > > | :--- | :--- | :--- | :--- | :--- | :--- |
> > > | Python | 543 | 0.585 | 0.419 | 0.245 | 133 |
> > > | Go | 502 | 0.820 | 0.552 | 0.453 | 227 |
> > > | Java | 47 | 0.464 | 0.794 | 0.369 | 17 |
> > > | JavaScript | 694 | 0.841 | 0.568 | 0.478 | 332 |
> > > | C | 16 | 1.000 | 0.582 | 0.582 | 9 |
> > > | Rust | 769 | 0.576 | 0.600 | 0.345 | 266 |
> > > | TypeScript | 157 | 0.719 | 0.260 | 0.187 | 29 |
> > > | PHP | 395 | 0.811 | 0.416 | 0.337 | 133 |
> > > | Ruby | 749 | 0.796 | 0.280 | 0.223 | 167 |
> > > | **Total** | **3,872** | **-** | **-** | **-** | **1,313** |
> > >
> > > Due to time constraints, the Supervised Fine-Tuning (SFT) training results will be updated in the revised version. However, given the substantial disparity in generated data volume between the two methods, the performance advantages of our approach are predictable.

---

### Official Review · Reviewer_5MhC · 2026-03-12

**Soundness:** 4
**Presentation:** 4
**Significance:** 4
**Originality:** 3
**Overall Recommendation:** 5
**Confidence:** 4

**Summary:**

This work addresses a notable aspect of automated software engineering datasets, specifically the challenge of constructing verifiable, multi-language executable environments at scale. The authors introduce MEnvAgent, a multi-agent framework that combines Planning, Execution, and Verification loops with an Environment Reuse Mechanism to efficiently build and validate Docker-based environments from real-world GitHub repositories. Evaluations on MEnvBench, a newly curated benchmark with 1,000 tasks across 10 programming languages, demonstrate improvements in Fail-to-Pass (F2P) rates by 8.6% and reductions in time costs by 43% compared to strong baselines. Additionally, the framework produces MEnvData-SWE, a large, open-source polyglot dataset with verified solution trajectories suitable for training and fine-tuning LLMs for software engineering tasks.

**Compliance With Llm Reviewing Policy:**

Affirmed.

**Final Justification:**

All of my concerns have been adequately addressed. I will keep my score and vote for acceptance.

**Key Questions For Authors:**

1. How sensitive is MEnvAgent’s performance to the size and diversity of the historical environment pool, and what minimal dataset size would be required for comparable gains?
2. Could the EnvPatch mechanism handle repository snapshots with incompatible version constraints or highly fragmented dependency graphs, and how often does patching fail in practice?
3. Are there limitations in scaling MEnvAgent to less common programming languages or domain-specific frameworks, and how would you extend support to these cases?
4. Can the framework accommodate dynamic repositories that change frequently, and how does it handle potential drift in dependency resolution or test behavior?

**Limitations:**

yes

**Strengths And Weaknesses:**

## Strengths
- The multi-agent Planning-Execution-Verification architecture effectively isolates responsibilities, allowing the Repository Analysis, Environment Setup, Test Configuration, Execution, Verification, and EnvPatch agents to collaboratively diagnose and resolve build failures, as illustrated in the diagram on page 3.
- A notable domain addressed by this article is polyglot environment construction; tasks span 10 programming languages and demonstrate robustness across both modern languages like Python and Go and complex ecosystems like Java and C/C++.
- The Environment Reuse Mechanism is incremental and context-aware, leveraging historical environments to reduce redundant computation while maintaining correctness, as shown in the case study on page 17.
- MEnvBench is carefully designed with inter-project diversity and intra-project depth, covering a wide range of domains and project scales, ensuring evaluation comprehensiveness.
- Empirical results show that MEnvAgent achieves high Fail-to-Pass rates with lower runtime, indicating simultaneous gains in efficiency and environment validity.
- Experiments scaling the number of instances per repository show that larger historical pools increase Reuse Success Rate and Pass Rate while lowering time costs, demonstrating practical scalability.
- Fine-tuning LLMs on MEnvData-SWE yields consistent performance improvements across benchmarks, confirming the utility of the constructed environments for downstream tasks.

## Weaknesses
- Handling less common or legacy language versions is not fully discussed, which may affect generalization to older software ecosystems.
- The human baseline for task validation is not extensively described; it remains unclear how closely the automated Fail-to-Pass verification aligns with expert judgment or whether corner cases might be missed.
- Evaluation focuses on standard open-source repositories; more challenging or proprietary repositories may introduce dependencies that could compromise the EnvPatch effectiveness.

---

> ### Author Rebuttal · Authors · 2026-03-31
>
> We thank the reviewer for their insightful comments. We will incorporate the suggestions in the revision.
>
> > **W1 & Q3: Handling less common or legacy language versions / Limitations in scaling to less common programming languages or domain-specific frameworks.**
>
> We thank the reviewer for highlighting this important frontier. Historically, ML and SWE agent research has been heavily Python-centric. By supporting 10 languages (e.g., Java, PHP, Rust), MEnvAgent significantly broadens language inclusion.
>
> Our framework is inherently **language-agnostic**. By utilizing standard command-line tools rather than language-specific plugins, we can seamlessly adapt to new languages without structural barriers. This adaptation simply requires providing appropriate base images and language-specific prompt contexts.
>
> The primary scaling limitation lies in the knowledge of the underlying LLM. If a model has poor zero-shot knowledge of a legacy build system, its success in drafting setup scripts or diagnosing errors will naturally be lower. While MEnvAgent is structurally ready for these languages, performance will scale with the base models' evolving understanding of these ecosystems.
>
> > **W2: The human baseline for task validation; alignment of automated Fail-to-Pass verification with expert judgment.**
>
> As detailed in our response to **Reviewer DjA9 (Q1)**, F2P serves as a reliable indicator for environment verification.
>
> This reliability is supported by prior work: **SWE-Factory** demonstrated **100% consistency** between F2P outcomes and human expert judgment across 2,085 instances. Our own manual evaluation of 360 instances from MEnvBench yielded the same result, demonstrating strict alignment with human assessment.
>
> Furthermore, our LLM-based Task Quality Assurance (DeepSeek-V3) demonstrates strong alignment when validated against the human-labeled ground truth from OpenAI’s SWE-bench Verified set:
>
> | Metric | Score |
> | :--- | :--- |
> | **Precision** | 80.37% |
> | **Recall** | 81.32% |
> | **F1 Score** | 80.84% |
>
> > **W3: Limitation to open-source repositories (lack of proprietary codebases).**
>
> We appreciate this observation and agree this is a genuine limitation. However, accessing proprietary codebases is an industry-wide challenge constrained by legal and licensing barriers, affecting all SWE research (e.g., SWE-bench, SWE-Factory). Given these constraints, open-source repositories currently provide the most practical and reproducible foundation for scalable research.
>
> We will explicitly add a discussion regarding this limitation in our revision. We hope future industry collaborations can extend verifiable environment construction to closed-source settings.
>
> > **Q1: How sensitive is MEnvAgent’s performance to the size and diversity of the historical environment pool, and what minimal dataset size would be required?**
>
> We investigated this directly in our ablation study, with results illustrated in Figure 4. MEnvAgent's performance is sensitive to the pool size, as the Reuse Success Rate (RSR) naturally scales with the availability of historical data. However, the minimal dataset size required for comparable gains is surprisingly low. As shown in Figure 4, even with a minimal scale of just 4 historical instances per repository, the framework achieves clear and significant improvements in both time efficiency and overall Pass Rate.
>
> > **Q2: Could the EnvPatch mechanism handle repository snapshots with incompatible version constraints or highly fragmented dependency graphs, and how often does patching fail in practice?**
>
> Our empirical results on MEnvBench show that **50.6% of all successful builds** were achieved through environment reuse  mechanism. This demonstrates its ability to resolve incompatible version constraints and fragmented dependency graphs in practice.
>
> For cases where patching is infeasible due to extreme conflicts, MEnvAgent gracefully aborts the attempt and falls back to building the environment from scratch. This hybrid approach ensures build reliability even when reuse is impossible.
>
> > **Q4: Can the framework accommodate dynamic repositories that change frequently, and how does it handle potential drift in dependency resolution or test behavior?**
>
> First, once successfully constructed, MEnvAgent persists environments as Docker images, isolating them from future repository updates. Furthermore, if your concern pertains to potential drift during the construction process, such as an outdated `requirements.txt` conflicting with newer codebase changes, MEnvAgent resolves this through its multi-round feedback mechanism. For example, if an installed package lacks a required API, the resulting test failure prompts the agent to analyze the error logs and autonomously adjust the dependency version to resolve the conflict.
>
> Thank you again for your valuable suggestions, and we look forward to further discussions with you.

---

> > ### Author Rebuttal · Reviewer_5MhC · 2026-04-03
> >
> > All of my concerns have been adequately addressed. I will keep my score and vote for acceptance.

---

### Official Review · Reviewer_DjA9 · 2026-03-13

**Soundness:** 3
**Presentation:** 4
**Significance:** 3
**Originality:** 3
**Overall Recommendation:** 5
**Confidence:** 4

**Summary:**

The paper describes MEnvAgent, a multi-language framework for automating environment construction via a multi-agent Planning-Execution-Verification architecture. The paper introduces MEnvBench, a benchmark for the task of environment construction. The results show high Fail-to-Pass rates.

**Compliance With Llm Reviewing Policy:**

Affirmed.

**Key Questions For Authors:**

1. What kind of verification methods have you incorporated ?
2. Have you looked into adoption of Module of Thought process, tying the mutli-agent environment to some particular development process ?
3. Multi-agent systems often are cast as distributed optimization, while the software process research area have been looking into methods to optimize the process (e.g. preceding work on process languages and enactment environment such as Little-JIL, trying to achieve the optimized stage of the Capability Maturity Model). In the AI era it is possible to such process optimization with greater ease.
You mentioned minimization of adaptation effort. Can you cast this effort as distributed optimization in a multi-agent system ?

**Limitations:**

Yes

**Strengths And Weaknesses:**

The approach is sound, it provides the automation of an environment construction. Compared to related work the MEnvAgent shows higher Fail-To-Pass rates and is multi-language. The paper is well-written. It contains the results of extensive experimentation and a case study in the Appendix. The experiment is well designed.

There are other systems with a similar purpose. This system differs in its multi-language ability.

---

> ### Author Rebuttal · Authors · 2026-03-31
>
> We thank the reviewer for their insightful comments and constructive feedback. We provide the following clarifications and discussions.
>
> > **Q1: What kind of verification methods have you incorporated?**
>
> We explain our verification methods from two distinct perspectives:
>
> **1. Verifying the Executable Environment:** To verify the validity of the constructed environment itself, we rely on **Execution-level Verification (Fail-to-Pass, F2P)**. Achieving F2P requires the entire build process (base image, dependency installation, test configuration) to be correctly configured; a single misconfigured component results in a systemic execution failure rather than a valid F2P outcome.
>
> **2. Verifying the Task Instances:** As highlighted by OpenAI's experience with SWE-bench Verified, F2P alone can be insufficient, as instances may have overly narrow tests or underspecified problems. To guarantee the validity of the instances themselves, we employ two additional steps:
> * **Task Quality Assurance:** An LLM evaluator (detailed in Appendix D.2) strictly filters out underspecified or unsolvable issues by discarding instances scoring below 5/10.
> * **Empirical Verification:** Fine-tuning on our verified trajectories yields significant performance gains across five distinct model families (Table 4), empirically proving the high quality and downstream verifiability of the instances.
>
> We acknowledge that no automated pipeline fully replaces human review, but our layered approach provides practical, robust quality assurance at scale (10 languages, 3,000+ instances) where manual annotation is infeasible.
>
> > **Q2: Adoption of the Module of Thought (MoT) process and tying the multi-agent environment to specific development processes.**
>
> Thank you for the valuable suggestion. During the early stages of our research, we surveyed various agent paradigms and concluded that a workflow-style design is most suitable for the unique requirements of software environment construction. Consequently, we adopted a three-stage architecture and introduced an environment reuse mechanism to maximize efficiency.
>
> While we did not explicitly formalize our framework under the Module of Thought (MoT) label, our current sub-agents for repository analysis, installation script generation, and test validation can be viewed as a form of modularized reasoning. We agree that these modules could be further decomposed and tied to more granular development processes via MoT. This represents a highly promising direction to further enhance the system's reasoning depth, and we will explore this refinement in our future work.
>
> > **Q3: Casting the minimization of adaptation effort as distributed process optimization within a multi-agent system.**
>
> Thank you for bringing Little-JIL to our attention. We were not previously familiar with this process programming language. Upon reviewing its design principles, we note that Little-JIL was developed within the paradigm of traditional software process engineering, where the focus was on coordinating predefined steps with formal control flow semantics. In the current era of LLM-based agents, the execution paradigm has shifted considerably; agents now exhibit greater autonomy, and coordination patterns tend to be more emergent rather than statically prescribed.
>
> That said, we find the reviewer's broader point regarding process optimization in multi-agent systems to be thought-provoking. We will investigate whether certain concepts from process programming—such as formal coordination structures and verifiable process properties—could be adapted to inform the design of our agent workflows in future iterations. We appreciate this suggestion and have added it as a direction for future work.
>
> Thank you again for your valuable suggestions, and we look forward to further discussions with you.

---

> > ### Author Rebuttal · Reviewer_DjA9 · 2026-04-04
> >
> > Thank you for clarifying my questions. Indeed, introducing process optimization into your multi-agent system is an interesting direction of future research.

---

### Decision · Program_Chairs · 2026-04-30

**Decision:**

Accept (spotlight)

**Comment:**

The authors proposed a multi-agent framework (called MEnvAgent) that utilizes a novel environment reuse mechanism to automate the scalable, multi-language construction of verifiable software execution environments. All reviewers agreed on its practical significance and praised the capability to efficiently handle repositories with strict, high Fail-to-Pass validation rates. While some reviewers raised concerns that the core novelty of this paper is more like the robust systems engineering rather than foundational algorithms, some other reviewers also mentioned that the open-source artifacts (including MEnvBench and MEnvData-SWE) provide substantial values to the diverse communities. During the rebuttal, the authors successfully addressed most concerns by providing a comprehensive ablation study that validated the reuse mechanism beyond Python as well as a detailed failure analysis for complex compiled languages. The robust empirical results, significant reductions in computational overhead and clear practical utility outweigh some limitations that the reviewers raised. Therefore, I recommend to accept this paper.